# Subquadratic Algorithms for Kernel Matrices via Kernel Density Estimation

**Ainesh Bakshi**
MIT
ainesh@mit.edu

**Piotr Indyk**
MIT
indyk@mit.edu

**Praneeth Kacham**
CMU
pkacham@cs.cmu.edu

**Sandeep Silwal**
MIT
silwal@mit.edu

**Samson Zhou**
UC Berkeley and Rice University
samsonzhou@gmail.com

## Abstract

Kernel matrices, as well as weighted graphs represented by them, are ubiquitous objects in machine learning, statistics and other related fields. The main drawback of using kernel methods (learning and inference using kernel matrices) is efficiency – given $n$ input points, most kernel-based algorithms need to materialize the full $n \times n$ kernel matrix before performing any subsequent computation, thus incurring $\Omega(n^2)$ runtime. Breaking this quadratic barrier for various problems has therefore, been a subject of extensive research efforts.

We break the quadratic barrier and obtain *subquadratic* time algorithms for several fundamental linear-algebraic and graph processing primitives, including approximating the top eigenvalue and eigenvector, spectral sparsification, solving linear systems, local clustering, low-rank approximation, arboricity estimation and counting weighted triangles. We build on the recently developed Kernel Density Estimation framework, which (after preprocessing in time subquadratic in $n$) can return estimates of row/column sums of the kernel matrix. In particular, we develop efficient reductions from *weighted vertex* and *weighted edge sampling* on kernel graphs, *simulating random walks* on kernel graphs, and *importance sampling* on matrices to Kernel Density Estimation and show that we can generate samples from these distributions in *sublinear* (in the support of the distribution) time. Our reductions are the central ingredient in each of our applications and we believe they may be of independent interest. We empirically demonstrate the efficacy of our algorithms on low-rank approximation (LRA) and spectral sparsification, where we observe a **9x** decrease in the number of kernel evaluations over baselines for LRA and a **41x** reduction in the graph size for spectral sparsification.

## 1 Introduction

For a kernel function $k : \mathbb{R}^d \times \mathbb{R}^d \to \mathbb{R}$ and a set $X = \{x_1 \ldots x_n\} \subset \mathbb{R}^d$ of $n$ points, the entries of the $n \times n$ kernel matrix $K$ are defined as $K_{i,j} = k(x_i, x_j)$. Alternatively, one can view $X$ as the vertex set of a complete weighted graph where the weights between points are defined by the kernel matrix $K$. Popular choices of kernel functions $k$ include the Gaussian kernel, the Laplace kernel, exponential kernel, etc; see (Schölkopf et al., 2002; Shawe-Taylor et al., 2004; Hofmann et al., 2008) for a comprehensive overview.

Despite their wide applicability, kernel methods suffer from drawbacks, one of the main being efficiency – given $n$ input points in $d$ dimensions, many kernel-based algorithms need to materialize the full $n \times n$ kernel matrix $K$ before performing the computation. For some problems this is unavoidable, especially if high-precision results are required (Backurs et al., 2017). In this work, we show that we can in fact break this $\Omega(n^2)$ barrier for several fundamental problems in numerical linear algebra and graph processing. We obtain algorithms that run in $o(n^2)$ time and scale inversely-proportional to the smallest entry of the kernel matrix. This allows us to skirt several known lower

bounds, where the hard instances require the smallest kernel entry to be polynomially small in $n$. Our parameterization in terms of the smallest entry is motivated by the fact in practice, the smallest kernel value is often a fixed constant (March et al., 2015; Siminelakis et al., 2019; Backurs et al., 2019; 2021; Karppa et al., 2022). We build on recently developed fast approximate algorithms for Kernel Density Estimation (Charikar & Siminelakis, 2017; Backurs et al., 2018; Siminelakis et al., 2019; Backurs et al., 2019; Charikar et al., 2020). Specifically, these papers present fast approximate data structures with the following functionality:

**Definition 1.1** (Kernel Density Estimation (KDE) Queries). *For a given dataset $X \subset \mathbb{R}^d$ of size $n$, kernel function $k$, and precision parameter $\varepsilon > 0$, a KDE data structure supports the following operation: given a query $y \in \mathbb{R}^d$, return a value $\mathrm{KDE}_X(y)$ that lies in the interval $[(1 - \varepsilon)z, (1 + \varepsilon)z]$, where $z = \sum_{x \in X} k(x, y)$, assuming that $k(x, y) \geq \tau$ for all $x \in X$.*

The performance of the state of the art algorithms for KDE also scales proportional to the smallest kernel value of the dataset (see Table 1). In short, after a preprocessing time that is sub-quadratic (in $n$), KDE data structures use time sublinear in $n$ to answer queries defined as above. Note that for all of our kernels, $k(x, y) \leq 1$ for all inputs $x, y$.

Table 1: Instantiations of KDE queries. The query times depend on the dimension $d$, accuracy $\varepsilon$, and lower bound $\tau$. The parameter $\beta$ is assumed to be a constant.

| Type | $k(x, y)$ | Preprocessing Time | Query Time | Reference |
|---|---|---|---|---|
| Gaussian | $e^{-\|x-y\|_2^2}$ | $\frac{nd}{\varepsilon^2 \tau^{0.173+o(1)}}$ | $\frac{d}{\varepsilon^2 \tau^{0.173+o(1)}}$ | (Charikar et al., 2020) |
| Exponential | $e^{-\|x-y\|_2}$ | $\frac{nd}{\varepsilon^2 \tau^{0.1+o(1)}}$ | $\frac{d}{\varepsilon^2 \tau^{0.1+o(1)}}$ | (Charikar et al., 2020) |
| Laplacian | $e^{-\|x-y\|_1}$ | $\frac{nd}{\varepsilon^2 \tau^{0.5}}$ | $\frac{d}{\varepsilon^2 \tau^{0.5}}$ | (Backurs et al., 2019) |
| Rational Quadratic | $\frac{1}{(1+\|x-y\|_2^2)^{\beta}}$ | $\frac{nd}{\varepsilon^2}$ | $\frac{d}{\varepsilon^2}$ | (Backurs et al., 2018) |

## 1.1 Our Results

We show that given a KDE data structure as described above, it is possible to solve a variety of matrix and graph problems in subquadratic time $o(n^2)$, i.e., sublinear in the matrix size. We emphasize that in our applications, we only require *black-box* access to KDE queries. Given this, we design algorithms for problems such as eigenvalue/eigenvector estimation, low-rank approximation, graph sparsification, local clustering, aboricity estimation, and estimating the total weight of triangles.

Our results are obtained via the following two-pronged approach. First, we use KDE data structures to design algorithms for the following basic primitives, frequently used in sublinear time algorithms and property testing:

1. sampling vertices by their (weighted) degree in $K$ (Theorems C.2 and C.4 and Algorithms 2 / 4),

2. sampling random neighbors of a given vertex by edge weights in $K$ and sampling a random weighted edge (Theorem C.5 and Algorithms 5 and 6),

3. performing random walks in the graph $K$ (Theorem C.7 and Algorithm 7), and

4. sampling the rows of the edge-vertex incident matrix and the kernel matrix $K$, both with probability proportional to respective row norms squared (Section D.1, Theorem D.1, and Section D.2, Corollary D.10 respectively).

In the second step, we use these primitives to implement a host of algorithms for the aforementioned problems. We emphasize that these primitives are used in a black-box manner, meaning that any further improvements to their running times will automatically translate into improved algorithms for the downstream problems. For our applications, we make the following parameterization, which we expand upon in Remark B.1 and Section B.1. At a high level, many of our applications, such as spectral sparsification, are succinctly characterized by the following parameterization.

**Parameterization 1.1.** *All of our algorithms are parameterized by the smallest edge weight in the kernel matrix, i.e., the smallest edge weight in the matrix $K$ is at least $\tau$.*

Table 2: Summary of linear algebra and graph applications for KDE subroutines. We suppress dependence on the precision $\varepsilon$. In spectral/local clustering and low-rank approximation, $k$ denotes the number of clusters and the rank of the approximation desired, respectively. The parameter $\phi$ refers to the quality of the underlying clusters; see Section E.1.

| Problem | # of KDE Queries | Post-processing time | Prior Work |
|---|---|---|---|
| Spectral sparsification (Thm. 1.2) | $\widetilde{O}\left(\frac{n}{\tau^3}\right)$ | $O\left(\frac{nd}{\tau^3}\right)$ | Remark B.1 |
| Laplacian system solver (Thm. 1.2) | $\widetilde{O}\left(\frac{n}{\tau^3}\right)$ | $O\left(\frac{nd}{\tau^3}\right)$ | Remark B.1 |
| Low-rank approx. (Thm. 1.5) | $O(n)$ | $O\left(n \cdot \text{poly}\left(k\right) + nkd\right)$ | Remark B.3 |
| Eigenvalue Spectrum approx. (Thm. 1.3) | $\widetilde{O}(1/\tau)$ | $O(d/\tau)$ | $\Omega(n^2 d)$ |
| Approximating 1st Eigenvalue (Thm. 1.4) | Remark B.2 | $d \cdot \text{poly}(1/\tau)$ | $\omega(n)$ (Remark B.2) |
| Local clustering (Thm. 1.6) | $\widetilde{O}\left(\text{poly}(k) \cdot \frac{1}{\phi^2} \frac{\sqrt{n}}{\tau^{1.5}}\right)$ | $\widetilde{O}\left(\text{poly}(k) \cdot \frac{1}{\phi^2} \frac{\sqrt{n}}{\tau^{1.5}}\right)$ | Remark B.4 |
| Spectral clustering (Thm. E.7) | $\widetilde{O}\left(\frac{n}{\tau^2}\right)$ | $O\left(\frac{nd}{\tau^2}\right) + \widetilde{O}\left(nk\right)$ | Remark B.4 |
| Arboricity estimation (Thm. 1.8) | $\widetilde{O}\left(\frac{n}{\tau}\right)$ | $\widetilde{O}\left(\frac{n^2}{\tau}\right)$ | $\widetilde{O}(n^3) + O(n^2 d)$ |
| Triangle estimation (Thm. 1.9) | $\widetilde{O}\left(\frac{1}{\tau^3}\right)$ | $\widetilde{O}\left(\frac{1}{\tau^3}\right)$ | $\Omega(n^2 d)$ |

Our applications derived from the basic graph primitives above can be partitioned into two overlapping classes, linear-algebraic and graph theoretic results. Table 2 lists our applications along with the number of KDE queries required in addition to any post-processing time. We refer to the specific sections of the body listed below for full details. We note that in *all of our theorems below*, we assume access to a KDE data structure of Definition 1.1 with parameters $\varepsilon$ and $\tau$.

One of our main results is spectral sparsification of the kernel matrix $K$ interpreted as a weighted graph. In Section D.1, we compute a sparse subgraph whose associated matrix closely approximates that of the kernel matrix $K$. The most meaningful matrix to study for such a sparsification is the Laplacian matrix, defined as $D - K$ where $D$ is a diagonal matrix of vertex degrees. The Laplacian matrix encodes fundamental combinatorial properties of the underlying graph and has been well-studied for numerous applications, including sparsification; see (Merris, 1994; Batson et al., 2013; Spielman, 2016) for a survey of the Laplacian and its applications. Our result computes a sparse graph, with a number of edges that is linear in $n$, whose Laplacian matrix spectrally approximates the Laplacian matrix of the original graph $K$ under Parameterization 1.1.

**Theorem 1.2** (Informal; see Thm. D.1). *Let $L$ be the Laplacian matrix corresponding to the graph $K$. Then, for any $\varepsilon \in (0,1)$, there exists an algorithm that outputs a weighted graph $G'$ with only $m = O(n \log n/(\varepsilon^2 \tau^3))$ edges, such that with probability at least $9/10$, $(1-\varepsilon)L \preceq L_{G'} \preceq (1+\varepsilon)L$. The algorithm makes $\widetilde{O}(m)$ KDE queries and requires $\widetilde{O}(md)$ post-processing time.*

We compare our results with prior works in Remark B.1. We also show that Parameterization 1.1 is **inherent** for spectral sparsification. In particular, we use a hardness result from (Alman et al., 2020) to show that for the Gaussian kernel, under the strong exponential time hypothesis (Impagliazzo & Paturi, 2001), any algorithm that returns an $O(1)$-approximate spectral sparsifier with $O(n^{1.99})$ edges requires $\Omega\left(n \cdot 2^{\log(1/\tau)^{0.32}}\right)$ time (see Theorem D.4 for a formal statement). Obtaining the optimal dependence on $\tau$ remains an outstanding open question, even for Gaussian and Laplace kernels. Spectral sparsification has further downstream applications in solving Laplacian linear systems, which we present in Section D.1.1.

Continuing the theme of the Laplacian matrix, in Section D.3, we also obtain a succinct summary of the entire eigenvalue spectrum of the (normalized) Laplacian matrix using a total number of KDE queries independent of $n$, the size of the dataset. The error of the approximation is measured in terms of the earth mover distance (see Eq. (D.1)), or EMD, between the approximation and the true set of eigenvalues. Such a result has applications in determining whether an underlying graph can be modeled from a specific graph generative process (Cohen-Steiner et al., 2018).

**Theorem 1.3** (Informal; see Theorem D.11). *Let $\varepsilon \in (0,1)$ be the error parameter and $L$ be the normalized Laplacian of the kernel graph $K$. Let $\lambda_1 \geq \lambda_2 \ldots \geq \lambda_n$ be the eigenvalues of $L$ and let $\lambda$ be the resulting vector. Then, there exists an algorithm that uses $\widetilde{O}\left(\exp\left(1/\varepsilon^2\right)/\tau\right)$ KDE queries*

*and* $\exp\left(1/\varepsilon^2\right) \cdot d/\tau$ *post-processing time and outputs a vector* $\widetilde{\lambda}$ *such that with probability* $99/100$, $EMD\left(\lambda, \widetilde{\lambda}\right) \leq \varepsilon$.

Again to the best of our knowledge, all prior works for approximating the spectrum in EMD require constructing the full graph beforehand, and thus have runtime $\Omega(n^2 d)$. Next, we obtain truly sublinear time algorithms for approximating the top eigenvalue and eigenvector of the kernel matrix, a problem which was studied in (Backurs et al., 2021). Our result is the following theorem. Our bounds, and those of prior work, depend on the parameter $p$, which refers to the exponent of $\tau$ in the KDE query runtimes. For example for the Gaussian kernel, $p \approx 0.173$. See Table 1 for other kernels.

**Theorem 1.4** (Informal; see Theorem D.15). *Given an $n \times n$ kernel matrix $K$ that admits a KDE data-structure with query time $d/(\varepsilon^2 \tau^p)$ (Table 1), there exists an algorithm that outputs a unit vector $v$ such that $v^T K v \geq (1 - \varepsilon)\lambda_1(K)$ in time $\min\left(\tilde{O}(d/(\varepsilon^{4.5}\tau^4)), \tilde{O}(d/(\varepsilon^{9+6p}\tau^{2+2p}))\right)$, where $\lambda_1(K)$ denotes the largest eigenvalue of $K$.*

We discuss related works in Remark B.2. In summary, the best prior result of (Backurs et al., 2021) had a runtime of $\Omega(n^{1+p})$ whereas our bound has no dependence on $n$. Finally, our last linear-algebraic result is an additive-error low-rank approximation of the kernel matrix, presented in Section D.2.

**Theorem 1.5** (Informal; see Cor. D.10). *There exists an algorithm that outputs a rank-$r$ matrix $B$ such that $\|K - B\|_F^2 \leq \|K - K_r\|_F^2 + \varepsilon \|K\|_F^2$ with probability $99\%$ where $K_r$ is the optimal rank-$r$ approximation of $K$. It uses $n$ KDE queries and $O(n \cdot \text{poly}(r, 1/\varepsilon) + nrd/\varepsilon)$ post-processing time.*

We give detailed comparisons between our results and prior work in Remark B.3. As a summary, (Bakshi et al., 2020b) obtain a *relative* error approximation with a running time of $\widetilde{O}\left(nd\left(r/\varepsilon\right)^{\omega-1}\right)$, where $\omega$ denotes the matrix multiplication constant, whereas our running time is dominated by $O(nrd/\varepsilon)$ and we obtain only additive error guarantees. Nevertheless, the algorithm we obtain, which builds upon the sampling scheme of (Frieze et al., 2004), is a conceptually simpler algorithm than the algorithm of (Bakshi et al., 2020b) and easier to empirically evaluate. Indeed, we implement this algorithm in Section 2 and show that it is highly competitive to the SVD.

We now move onto graph applications. We obtain an algorithm for local clustering, where we are asked whether two vertices belong to the same or different vertex communities. The notion of a cluster structure is based on the definition of a $k$-clusterable graph, formally introduced in Definition E.3. Intuitively, it describes a graph whose vertices can be partitioned into $k$ disjoint clusters with high-connectivity within clusters and relatively sparse connectivity in-between clusters.

**Theorem 1.6** (Informal; see Theorem E.5). *Let $K$ be a $k$-clusterable kernel graph with clusters $V = \cup_{1 \leq i \leq k} V_i$. Let $U, W$ be one of (not necessarily distinct) clusters $V_i$. Let $u, w$ be randomly chosen vertices in partitions $U$ and $W$ with probability proportional to their degrees. There exists $c = c(\varepsilon, k)$ and an algorithm that uses $\widetilde{O}(c(k, \varepsilon)\sqrt{n}/\tau^{1.5})$ KDE queries and post-processing time, with the property that with probability $1 - \varepsilon$, if $U = W$ then the algorithm reports that $u$ and $w$ are in the same cluster and if $U \neq W$, the algorithm reports that $u$ and $w$ are in different clusters.*

Our definitions for the local clustering result are adopted from prior literature in property testing; see Remark B.4 for an overview of related works. Our sparsification result also automatically lends itself to an application in spectral clustering, an algorithm that clusters vertices based on the eigenvectors of the Laplacian matrix, which is outlined in Section E.2. We obtain an algorithm for approximately computing the top few eigenvectors of the Laplacian matrix, which is one of the main bottlenecks in spectral clustering in practice, with subquadratic runtime. These approximate eigenvectors are used to form the clusters.

**Theorem 1.7** (Informal; see Theorem E.8). *Let $L$ be the Laplacian matrix of the spectral sparsifier. There exists an algorithm that can compute $(1 + \varepsilon)$-approximations of the first $k$ eigenvectors of $L$ in time $\widetilde{O}\left(kn/(\tau^2 \varepsilon^{2.5})\right)$.*

We also give algorithms for approximating the arboricity of a graph, which is the density of the densest subgraph of the kernel graph (see exact definition in Section E.3).

**Theorem 1.8** (Informal; see Theorem E.9). *There exists an algorithm that uses $m = \widetilde{O}(n/(\varepsilon^2 \tau))$ KDE queries and $O(mn)$ post-processing time and outputs a sparse subgraph $G'$ of the kernel graph such that with high probability, $(1 - \varepsilon)\alpha_G \leq \alpha_{G'} \leq (1 + \varepsilon)\alpha_G$, where $\alpha_G$ is the arboricity of $G$.*

To the best of our knowledge, all prior works on computing the arboricity require the entire graph to be known beforehand. In addition, computing the arboricity requires time $\widetilde{O}(nm)$ where $m$ is the number of edges leading to a runtime of $\widetilde{O}(n^3) + O(n^2 d)$ (Gallo et al., 1989). In Section E.4, we also give an algorithm for approximating the total weight of all triangles of $K$, again interpreted as a weighted graph. We define weight of a triangle as the product of its edge weights. This is a natural definition if weighted edges are interpreted as parallel unweighted edges, in addition to having applications in defining cluster coefficients of weighted graphs (Kalna & Higham, 2006; Li et al., 2007; Antoniou & Tsompa, 2008). Our bound is similar in spirit to the bound of the unweighted case given in (Eden et al., 2017), under a different computation model. We refer to Remark B.5 for discussions on related works.

**Theorem 1.9** (Informal; see Theorem E.10). *There exists an algorithm that makes $\widetilde{O}(1/\tau^3)$ KDE queries and the same bound for post-processing time and with probability at least $\frac{2}{3}$, outputs a $(1 \pm \varepsilon)$-approximation to the total weight of the triangles in the kernel graph.*

On the other hand, there is a line of work that considers dimensionality reduction for kernel density estimation e.g., through coresets (Phillips & Tai, 2018; 2020a; Tai, 2022). We view this direction of work as orthogonal to our line of study. Lastly, the work (Backurs et al., 2021) is similar in spirit to our work as they also utilize KDE queries to speed up algorithms for kernel matrices. Besides top eigenvalue estimation mentioned before, (Backurs et al., 2021) also study the problem of estimating the sum of all entries in the kernel matrix and obtain tight bounds for the latter.

## 1.2 Technical Overview

We provide a high-level overview and intuition for our algorithms. We first highlight our algorithmic building blocks for fundamental tasks and then describe how these components can be used to handle a wide range of problems. We note that our building blocks use KDE data structures in a black-box way and thus we describe their performance in terms of the number of queries to a KDE oracle. We also note that a permeating theme across all subsequent applications is that we want to perform some algorithmic task on a kernel matrix $K$ without computing each of its entries $k(x_i, x_j)$.

**Algorithmic Building Blocks.** We first describe the "multi-level" KDE data structure, which constructs a KDE data structure on the entire input dataset $X$, and then recursively partitions $X$ into two halves, building a KDE data structure on each half. The main observation here is that if the initialization of a KDE data structure uses runtime linear in the size $n$ of $X$, then at each recursive level, the initialization of the KDE data structures across all partitions remains linear. Since there are $O(\log n)$ levels, the overall runtime to initialize our multi-level KDE data structure incurs only a logarithmic overhead (see Figure 1 for an illustration).

**Weighted vertex sampling.** We describe how to sample vertices approximately proportional to their weighted degree, where the weighted degree of a vertex $x_i$ with $i \in [n]$ is $w_i = \sum_{j \neq i} k(x_i, x_j)$. We observe that performing $n$ KDE queries suffices to get an approximation of the weighted vertex degree of all $n$ vertices. We can thus think of vertex sampling as a preprocessing step that uses $n$ queries upfront and then allows for arbitrary sample access at any point in the future with no query cost. Moreover, this preprocessing step of taking $n$ queries only needs to be performed once. Further, we can then perform weighted vertex sampling from a distribution that is $\varepsilon$-close in total variation to the true distribution (see Theorem C.4 for details). Here, we use a multi-level tree structure to iteratively choose a subset of vertices with probability proportional to its approximate sum of weighted degrees determined by the preprocessing step, until the final vertex is sampled. Hence after the initial $n$ KDE queries, each query only uses $O(\log n)$ runtime, which is significantly better than the naïve implementation that uses quadratic time to compute the entire kernel matrix.

**Weighted neighbor edge sampling.** We describe how to perform weighted neighbor edge sampling for a given vertex $x$. The goal of weighted neighbor edge sampling is to efficiently output a vertex $v$ such that $\Pr[v = x_k] = \frac{(1 \pm \varepsilon)k(x, x_k)}{\sum_{j \in [n], x_j \neq x} k(x, x_j)}$ for all $k \in [n]$. Unlike the degree case, edge sampling

is not a straightforward KDE query since the sampling probability is proportional to the kernel value between two points, rather than the sum of multiple kernel values that a KDE query provides. However, we can utilize a similar tree procedure as in Figure 1 in conjunction with KDE queries.

In particular, consider the tree in Figure 1 where each internal node corresponds to a subset of neighbors of $x$. The two children of a parent node in the tree are simply the two approximately equal subsets whose union make up the subset representing the parent node. We can descend down the tree using the same probabilistic procedure as in the vertex sampling case: at every node, we pick one of the children to descend into with probability proportional to the sum of the edge weights represented by the children. The sum of edge weights of the children can be approximated by a query to an appropriate KDE data structure in the "multi-level" KDE data structure described previously. By appropriately decreasing the error of KDE data structures at each level of the tree, the sampled neighbor satisfies the aforementioned sampling guarantee. Since the tree has height $O(\log n)$, then we can perform weighted neighbor edge sampling, up to a tunably small total variation distance, using $O(\log n)$ KDE queries and $O(\log n)$ time (see theorems C.5 and C.6 for details).

**Random walks.** We use our edge sampling procedure to output a random walk on the kernel graph, where at any current vertex $v$ of the walk, the next neighbor of $v$ visited by the random walk is chosen with probability proportional to the edge weights adjacent to $v$. In particular, for a random walk with $T$ steps, we can simply sequentially call our edge sampling procedure $T$ times, with each instance corresponding to a separate step in the random walk. Thus we can perform $T$ steps of a random walk, again up to a tunably small total variation distance, using $O(T \log n)$ KDE queries and $O(T \log n)$ additional time.

**Importance Sampling for the edge-vertex incidence matrix and the kernel matrix.** We now describe how to sample the rows of the edge vertex incident matrix $H$ and the kernel matrix $K$ with probability proportional to the importance sampling score / leverage score (see Definition D.2). We remark that approximately sampling proportional to the *leverage score* distribution for $H$ is a fundamental algorithmic primitive in spectral graph theory and numerical linear algebra. We note that apriori, such a task seems impossible to perform in $o(n^2)$ time, even if the leverage scores are precomputed for us, since the support of the distribution has size $\Theta(n^2)$. However, note we do not need to compute (even approximately) each leverage score to perform the sampling, but rather just output an edge proportional to the right distribution.

We accomplish this by instead sampling proportional to the squared Euclidean norm of the rows of $H$. It is known that oversampling the rows of a matrix by a factor that depends on the condition number is sufficient to approximate leverage score sampling (see proof of Theorem D.1). Further, we show that $H$ has a condition number (Lemma D.3) that is bounded by $\mathrm{poly}(1/\tau)$. Recall, the edge-vertex incident matrix is defined as the $\binom{n}{2} \times n$ matrix with the rows indexed by all possible edges and the columns indexed by vertices. For each $e = \{i, j\}$, we have $H_{\{i,j\},i} = \sqrt{k(x_i, x_j)}$ and $H_{\{i,j\},j} = -\sqrt{k(x_i, x_j)}$. We pick the ordering of $i$ and $j$ arbitrarily. Note that this is a weighted analogue of the standard edge-vertex incident matrix and satisfies $H^T H = L_G$ where $L_G$ is the Laplacian matrix of the graph corresponding to the kernel matrix $K$. For both $H$ and $K$, we wish to sample the rows with probability proportional to row normed squared. For example, the row $r_e$ corresponding to edge $e = (x_i, x_j)$ in $H$ satisfies $\|r_e\|_2^2 = 2k(x_i, x_j)$. Since the squared norm of each row is proportional to the weight of the corresponding edge, we can perform this sampling by combining the weighted vertex sampling and weighted neighbor edge sampling primitives: we first sample a vertex with probability proportional to its degree and then sample an appropriate random neighbor. Thus our row norm sampling procedure is sufficient to simulate leverage score sampling (up to a condition number factor), which implies our downstream application of spectral sparsification.

We now describe the related primitive of sampling the rows of the kernel matrix $K$. Naïvely performing this sampling would require us to implicitly compute the entire kernel matrix, which as mentioned previously, is prohibitive. However, if there exists a constant $c$ such that the kernel function $k$ that defines the matrix $K$ satisfies $k(x, y)^2 = k(cx, cy)$ for all inputs $x, y$, then the $\ell_2^2$ norm of each row can be approximated via a KDE query on the transformed dataset $X' = cX$. In particular, the $\ell_2^2$ row norms of $K$ are the vertex degrees of the kernel graph for $X'$. The property that $k(x, y)^2 = k(cx, cy)$ holds for the most popular kernels such as the Laplacian, exponential, and Gaussian kernels. Thus, we can sample the rows of the kernel matrix with the desired probabilities.

**Linear Algebra Applications.** We now discuss our linear algebra applications.

**Spectral sparsification.** Using the previously described primitives of weighted vertex sampling and weighted neighbor edge sampling, we show that an $\varepsilon$ spectral sparsifier for the kernel density graph $G$ can be computed i.e., we compute a graph $G'$ such that for all vectors $x$, $(1 - \varepsilon)x^T L_G x \leq x^T L_{G'} x \leq (1 + \varepsilon)x^T L_{G'} x$, where $L_G$ and $L_{G'}$ denote the Laplacian matrices of the graphs $G$ and $G'$. Recall that $H$ is the $\binom{n}{2} \times n$ matrix such that $H_{\{i,j\},i} = \sqrt{k(x_i, x_j)}$ and $H_{\{i,j\},j} = -\sqrt{k(x_i, x_j)}$. Here we use subsets of $[n]$ of size 2 to index the rows of $H$ and the entry to be made negative in the above definition is picked arbitrarily. It can be verified that $H^T H = L_G$. It is known that sampling $t = O(n \log(n)/\varepsilon^2)$ rows of the matrix $H$ by using the so-called leverage scores gives a $t \times \binom{n}{2}$ selecting-and-scaling matrix $S$ such that with probability at least $9/10$,

$$(1 - \varepsilon)L_G = (1 - \varepsilon)H^T H \preceq H^T S^T S H \preceq (1 + \varepsilon)H^T H = (1 + \varepsilon)L_G. \tag{1.1}$$

Thus the matrix $SH$ directly corresponds to a graph $G'$, which is an $\varepsilon$ spectral sparsifier for graph $G$. The leverage scores of rows of $H$ are also called "effective resistances" of edges of graph $G$. Unfortunately, with the edge and neighbor vertex sampling primitives that we have, we cannot perform leverage score sampling of $H$. On the other hand, observe that the squared norm of row $\{i, j\}$ of $H$ is $2k(x_i, x_j)$ and with an application of vertex sampling and edge sampling, we can sample a row of $H$ from the length squared distribution i.e., the distribution on rows where probability of sampling a row is proportional to its squared norm. It is a standard result that sampling from squared length distribution gives a selecting-and-scaling matrix $S$ that satisfies (1.1), although we have to sample $t = O(\kappa^2 n \log(n)/\varepsilon^2)$ rows from this distribution, where $\kappa = \sigma_{\max}(H)/\sigma_{\min}(H)$ denotes the condition number of $H$ ($\sigma_{\max}(H)$ (resp. $\sigma_{\min}(H)$) denotes the largest (resp. smallest) *positive* singular values).

With the parameterization that for all $i \neq j$, $k(x_i, x_j) \geq \tau$, we are able to show that $\kappa \leq O(1/\tau^{1.5})$. Importantly, our upper bound on the condition number is independent of the data dimension and number of input points. We obtain the upper bound on condition number by using a Cheeger-type inequality for weighted graphs. Note that $\sigma_{\min}(H) \geq \sqrt{\lambda_2(H^T H)} = \sqrt{\lambda_2(L_G)}$, where we use $\lambda_2(M)$ to denote the second smallest eigenvalue of a positive semidefinite matrix. Cheeger's inequality lower bounds exactly the quantity $\lambda_2(L_G)$ in terms of graph conductance. A lower bound of $\tau$ on every kernel value implies that every node in the Kernel Graph has a high weighted degree and this lets us lower bound $\lambda_2(G)$ in terms of $\tau$ using a Cheeger-type inequality from (Friedland & Nabben, 2002) and shows that $O(n \log(n)/\tau^3 \varepsilon^2)$ samples from the approximate squared length sampling distribution gives an $\varepsilon$ spectral sparsifier for the graph $G$.

**First eigenvalue and eigenvector approximation.** Our goal is to compute a $1 - \varepsilon$ approximation to $\lambda$, the first eigenvalue of $K$, and an accompanying approximate eigenvector. Such a task is key in kernel PCA and related methods. We begin by noting that under the natural constraint that each row of $K$ sums to at least $n\tau$, a condition used in prior works (Backurs et al., 2021), the first eigenvalue must be at least $n\tau$ by looking at the quadratic form associated with the all-ones vector.

Now we combine two disparate families of algorithms: first the guarantees of (Bhattacharjee et al., 2021; Bakshi et al., 2020a) show that sub-sampling a $t \times t$ principal submatrix of a PSD matrix preserves the eigenvalues of the matrix up to an additive $O(n/\sqrt{t})$ factor. Since we've shown the first eigenvalue of $K$ is at least $n\tau$, we can set $t$ roughly $O(1/(\varepsilon^2 \tau^2))$ with the guarantee that the top eigenvalue of the sub-sampled matrix is at lest $(1 - \varepsilon)\lambda$. Now we can either run the standard Krylov method algorithm (Musco & Musco, 2015) to compute the top eigenvalue of the sampled matrix or alternatively, we can instead use the algorithm of (Backurs et al., 2021), the prior state of the art, to compute the eigenvalues of the sampled matrix. At a high level, their algorithm utilizes KDE queries to approximately perform power method on the kernel graph without creating the kernel matrix. In our case, we can instead run their algorithm on the smaller sampled dataset, which represents a smaller kernel matrix. Our final runtime is independent of $n$, the size of the dataset, whereas the prior state of the art result of (Backurs et al., 2021) have a $\omega(n)$ runtime.

**Graph Applications.** We now discuss our graph applications.

**Local clustering.** The random walks primitive allow us to run a well-studied local clustering algorithm on the kernel graph. The algorithm is quite standard in the property testing literature (see (Czumaj et al., 2015) and (Peng, 2020)) so we see our main contribution here as showing how the algorithm can be initialized for kernel matrices using our building blocks. At a high level, the goal

of the algorithm is to determine if two input vertices $u$ and $v$ belong to the same cluster of the kernel graph if the graph has a natural cluster structure (see Definition E.3 for the formal definition). The well-studied algorithm in literature performs approximately $O(\sqrt{n})$ random walks from $u$ and $v$ of a logarithmic length which is sufficient to estimate the distance between the endpoint distribution of the random walks. If the vertices belong to the same cluster, the distributions are close in $\ell_2$ distance which can be detected via a standard distribution tester of (Chan et al., 2014). The guarantees of the overall local clustering algorithm of (Czumaj et al., 2015) follow for kernel graphs since we only need to access the graph via random walks.

**Arboricity estimation.** The arboricity of a weighted graph $G = (V, E, w)$ is defined as $\alpha :=$ $\max_{U \subseteq V} \frac{w(E(G_U))}{|U|}$. Informally, the arboricity of a (weighted) graph represents the maximum (weighted) density of a subgraph of $G$. To approximate the weighted arboricity, we adapt a result of (McGregor et al., 2015), who observed that to estimate the arboricity on unweighted graphs, it suffices to sample a set of $\tilde{O}(|V|/\varepsilon^2)$ edges of $G$ and computes the arboricity of the subsampled graph, after rescaling the weight of edges inversely proportional to their sampling probabilities.

We show that a similar idea works for estimating arboricity on weighted graphs. Although (McGregor et al., 2015) showed that each edge should be sampled independently without replacement, we show that it suffices to sample a fixed number of edges with replacement. Moreover, we show that each edge should be one of the weighted edges with probability proportional to the weight of the edges, i.e., importance sampling. In fact, a similar result still holds if we only have upper bounds on the weight of each edge, provided that we increase the number of fixed edges that we sample by the gap between the upper bound and the actual weight of the edge. Thus, our arboricity algorithm requires sampling a fixed number of edges, where each edge is sampled with probability proportional to some known upper bound on its weight. However for kernel density graphs, this is just our weighted edge sampling subroutine. Therefore, we achieve improved runtime over the naïve approach of querying each edge in the kernel graph by using our weighted edge sampling subroutine to sample a fixed number of edges. Finally, we compute and output the arboricity of the subsampled graph as an approximation to the arboricity of the input graph.

## 2 Empirical Evaluation

We present empirical evaluations for our algorithms. We chose to evaluate algorithms for low-rank approximation (LRA) and spectral sparsification (and spectral clustering as a corollary) as they are arguably two of the most well studied examples in our applications and utilize a wide variety of techniques present in our other examples of Sections D and E. Our evaluations serve as a proof of concept that our queries which we constructed are efficient and easy to implement in practice. For our experiments, we use the Laplacian kernel $k(x, y) = \exp(-\|x - y\|_1/\sigma)$. A fast KDE implementation of this kernel exists due to (Backurs et al., 2019), which builds upon the techniques of (Charikar & Siminelakis, 2017). Note that the focus of our work is to use KDE queries in a mostly black box fashion to solve important algorithmic problems for kernel matrices. This viewpoint has the important advantage that it is flexible to the choice of any particular KDE query instantiation. We chose to work with the implementation of (Backurs et al., 2019) since it possesses theoretical guarantees, has an accessible implementation[1], and has been used in experiments in prior works such as (Backurs et al., 2019; 2021). However, we envision other choices of KDE queries, which maybe have practical benefits but are theoretically incomparable would also work well due to our flexibility.

**Datasets.** We use two real and two synthetic datasets in our experiments. The datasets used in the low-rank approximation experiments are MNIST (points in $\mathbb{R}^{784}$) (LeCun, 1998) and Glove word embeddings (points in $\mathbb{R}^{200}$) (Pennington et al., 2014). We use $10^4$ points from each of the test datasets. These datasets have been used in prior experimental works on kernel density estimation (Siminelakis et al., 2019; Backurs et al., 2019). The datasets in experimental results for spectral sparsification and clustering are described in detail in F.

**Evaluation Metrics.** For LRA, we use the additive error algorithm detailed in Corollary D.10 of Section D.2. It requires sampling the rows of the kernel matrix according to squared row norms,

---

[1]from `https://github.com/talwagner/efficient_kde`

which can be done via KDE queries as outlined there. Once the (small) number of rows are sampled, we explicitly construct these rows using kernel evaluations. We compare the approximation error of this method computed via the standard frobenius norm error to a state of the art sketching algorithm for computing low-rank approximations, which is the input-sparsity time algorithm of Clarkson and Woodruff (Clarkson & Woodruff, 2013) (**IS**). We also compare to an iterative SVD solver (**SVD**). All linear algebra subroutines rely on Numpy, Scipy, and Numba implementations when applicable.

**Low-rank approximation results.** Note that the algorithm in Corollary D.10 has a $O(k)$ dependence on the number of rows sampled. Concretely we sample $25k$ rows for a rank $k$ approximation which we fix it for all experiments. For the MNIST dataset, the rank versus approximation error is shown in Figure 2a. The performance of our algorithm labeled as **KDE** is given by the blue curve while the orange curve represents the **IS** algorithm. The green curve represents the SVD error, which is a lower bound on the error for any algorithm. Note that for SVD calculations, we do not calculate the full SVD since that is computationally prohibitive; instead, we use an iterative solver. We can see that the errors of all three methods are comparable to each other. In terms of runtime, the KDE based method took 24.7 seconds on average for the rank 50 approximation whereas **IS** took 71.5 seconds and iterative SVD took 74.72 seconds on average. This represents a **2.9x** decrease in the running time. The time measured includes the time to initialize the data structures and matrices used for the respective algorithms. In terms of the number of kernel evaluations, both **IS** and iterative SVD require the kernel matrix, which is $10^8$ kernel evaluations. On the other hand for the rank 50 approximation, our method required only $1.1 \cdot 10^7$ kernel evaluations, which is a **9x** decrease in the number of evaluations. In terms of space, **IS** and iterative SVD require $10^8$ floating point numbers stored due to initializing the full $10^4 \times 10^4$ matrix whereas our method only requires $10^4 \cdot 25 \cdot 50$ floating point numbers for the rank equal to 50 case and smaller for other. This is a **8x** decrease in the space required. Lastly, we verify that we are indeed sampling from the correct distribution required by Corollary D.10. In Figure 2b, we plot the points $(x_i, y_i)$ where $x_i$ is the row norm squared for the $i$th row of the kernel matrix $K$ and $y_i$ is the row norm squared computed in our approximation algorithm (see Algorithm 9). As shown in Figure 2b, the data points fall very close to the $y = x$ line indicating that our algorithm is indeed sampling from approximately the correct ideal distribution.

The qualitatively similar results for the Glove dataset are given in Figures 2c and 2d. For the glove dataset, the average time taken by the three algorithms were $37.7s, 37.7s$, and $44.2s$ respectively, indicating that **KDE** and **IS** were comparable in runtime whereas SVD took slightly longer. However, the number of kernel evaluations required by the latter two algorithms was significantly larger: for rank equal to 10, our algorithm only required $2.6 \cdot 10^6$ kernel evaluations while the other methods both required $10^8$ due to initializing the matrix. The space required by our algorithm was also smaller by a factor of 40 since we only explicitly compute $25 \cdot 10$ rows for the rank $= 10$ case. For Glove, we only perform our experiments up to rank 10 since the iterative SVD failed to converge for higher ranks. While computing the full SVD avoids the convergence issue, it's computationally prohibitive. For example for MNIST, computing the full SVD of the kernel matrix took $552.9s$, which is approximately an order of magnitude longer than any of the other methods.

**Acknowledgements** Ainesh Bakshi was supported by Ankur Moitra's ONR grant. Praneeth Kacham was supported by National Institute of Health (NIH) grant 5401 HG 10798-2, a Simons Investigator Award of David P. Woodruff, and Google as part of the "Research Collabs" program. Piotr Indyk and Sandeep Silwal were supported by the NSF TRIPODS program (award DMS-2022448), Simons Investigator Award, MIT-IBM Watson AI Lab and NSF Graduate Research Fellowship under Grant No. 1745302. Work done in part while Samson Zhou was at Carnegie Mellon University and supported by a Simons Investigator Award of David P. Woodruff and by the National Science Foundation under Grant No. CCF-1815840.

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

# A    Omitted Technical Overview

We provide a technical overview for applications whose discussions were omitted from the main text.

**Kernel matrix low-rank approximation.**    In this setting, our goal here is to output a matrix $B$ such that

$$\|K - B\|_F^2 \leq \|K - K_t\|_F^2 + \varepsilon \|K\|_F^2$$

where $K_t$ is the best rank-$t$ approximation to the kernel matrix $K$. The efficient algorithm of (Frieze et al., 2004) is able to achieve this guarantee if one can sample the $i$th row $r_i$ of $K$ with probability $p_i \geq \Omega(1) \cdot \|r_i\|_2^2 / \|K\|_F^2$. We can perform such an action using our primitive, which is capable of sampling the rows of $K$ with probability proportional to the squared row norms for the Laplacian, exponential, and Gaussian kernels. Thus for these kernels, we can immediately obtain efficient algorithms for computing a low-rank approximation.

**Spectrum approximation.**    For this problem, the goal is to compute approximations of all the eigenvalues of the normalized Laplacian matrix of the kernel graph such that the error between the approximations and the true set of eigenvalues has small error in the earth mover metric. The algorithm of (Cohen-Steiner et al., 2018) achieves this guarantee in time independent in the graph size given the ability to perform random walks on uniformly sampled vertices. Surprisingly, the number of random walks and their length does not depend on the number of vertices. Thus given our random walk primitive, we can efficiently simulate the algorithm of (Cohen-Steiner et al., 2018) on kernel graphs in a black-box manner.

**Spectral clustering.**    Given our spectral sparsification result, we can immediately obtain a fast version of a heuristic algorithm used in practice for graph clustering: we embed each vertex into $\mathbb{R}^k$ using $k$ eigenvectors of the Laplacian matrix and run $k$-means clustering. Clearly if we have a sparse graph, the eigenvector computation is faster. Theoretically, we can show that spectral sparsification preserves a notion of clusterability which is weaker than the definition used in the local clustering section and we additionally give empirical evidence of the validity of this procedure.

**Weighted triangle estimation.** We define the weight of a triangle as the product of its edges, generalizing the case were the edges have integer lengths, so that an edge can be thought of as multiple parallel edges. Under this definition, we adapt an algorithm from (Eden et al., 2017), who considered the problem in unweighted graphs given query access to the underlying graph. Specifically, we show that it suffices to sample a "small" set $R$ of edges uniformly at random and then estimate the total weight of triangles including the edges of $R$ under some predetermined ordering. In particular, the procedure of estimating the total weight of triangles including the edges $e$ of $R$ involves sampling neighbors of the vertices of $e$, which we can efficiently implement using our weighted neighbor edge sampling subroutine.

# B  Further Related Works

**Remark B.1.** Spectral sparsification for kernel graphs has also been studied in prior works, notably in (Alman et al., 2020) and (Quanrud, 2021). We first compare to (Alman et al., 2020), who obtain a spectral sparsification using an entirely different approach. They obtain an almost linear time sparsifier ($n^{1+o(1)}$) when the kernel is *multiplicativily Lipschitz* (see Section 1.1.2 in (Alman et al., 2020) for definition) and show hardness for constructing such a sparsifier when it is not. Focusing on the Gaussian kernel, under Parameterization 1.1, (Alman et al., 2020) obtain an algorithm that runs in time $O\left(nd + n2^{\sqrt{\log(1/\tau)\log n}\log(\log n)}/\varepsilon^2\right)$, whereas our algorithm runs in $O\left(nd\log^2(n)/(\varepsilon^2\tau^{2.0173+o(1)})\right)$ time. We also note that the dimension $d$ can be upper bounded by $O(\log n/\varepsilon^2)$ by applying Johnson-Lindenstrauss to the initial dataset. Therefore, (Alman et al., 2020) obtain a better dependence on $1/\tau$, whereas we obtain a better dependence on $n$. A similar comparison can be established for other kernels as well. In practice, $\tau$ is set to be a small fixed constant, whereas $n$ can be arbitrarily large. Indeed in practice, a common setting of $\tau$ is 0.01 or 0.001, irrespective of the size of the dataset (March et al., 2015; Siminelakis et al., 2019; Backurs et al., 2019; 2021; Karppa et al., 2022).

We now compare our guarantees to that of (Quanrud, 2021). The author studies spectral sparsification resurrected to *smooth* kernels (for example kernels of the form $1/\|x - y\|_2^t$ which have a polynomial decay; see (Quanrud, 2021) for a formal definition). This family *does not* include Gaussian, Laplacian, or exponential kernels. For smooth kernels, (Quanrud, 2021) obtained a sparsifier with a nearly optimal $\tilde{O}(n/\varepsilon^2)$ number of edges in time $\tilde{O}(nd/\varepsilon^2)$. Our algorithm obtains a similar dependence in $n, d, \varepsilon$ but includes an additional $1/\tau^3$ factor. However, it generalizes for any kernel supporting a KDE data structure, which includes smooth kernels (Backurs et al., 2018) (see Table 1 for a summary of kernels where our results apply). Our techniques are also different: (Quanrud, 2021) does not use KDE data structures in a black-box manner to compute the sparsification as we do. Rather, they simulate importance sampling on the edges of the kernel graph directly. In addition to the nearly linear sparsifier, another interesting feature of (Quanrud, 2021) is that it enriches the connections between spectral sparsification of kernel graphs and KDE data structures. Indeed, the data structures used in (Quanrud, 2021) are inspired by and were used in the prior work of (Backurs et al., 2018) to create KDE query data structures themselves. Furthermore, the paper demonstrates how to instantiate KDE data structures for smooth kernels using the kernel graph sparsifier itself. We refer to (Quanrud, 2021) for details.

**Remark B.2.** We remark that our algorithm returns a *sparse* vector $v$ supported on roughly $O(1/(\varepsilon^2\tau^2))$ coordinates. The best prior result is that of (Backurs et al., 2021) which presented an algorithm with total runtime $O\left(\frac{dn^{1+p}\log(n/\varepsilon)^{2+p}}{\varepsilon^{7+4p}}\right)$.

In comparison, our bound has *no dependence* on $n$ and is thus a *truly sublinear* runtime. Note that the bound of (Backurs et al., 2021) does not depend on $\tau$. We do not state the number of KDE queries used explicitly in Table 2 since our algorithm uses KDE queries on a subsampled dataset and in addition, only uses them by calling the algorithm of (Backurs et al., 2021) as a subroutine (on the subsampled dataset). The algorithm of (Backurs et al., 2021) uses $\tilde{O}(1/\varepsilon)$ KDE queries but with various different initialization of $\tau$ so it is not meaningful to state "one" bound for the number of KDE queries used and thus the final runtime is a more meaningful quantity to state. Lastly, the authors in (Backurs et al., 2021) present a lower bound of $\Omega(nd)$ for estimating the top eigenvalue $\lambda_1$, which ostensibly seems at odds with our stated bound which has no dependence on $n$. However,

the lower bound presented in (Backurs et al., 2021) essentially sets $\tau = 1/\operatorname{poly}(n)$ for a large polynomial factor depending on $n$ (we estimate this factor to be $\Omega(n^2)$). Since we parameterize our dependence via $\tau$, which in practice is often set to a fixed constant, we can bypass the lower bound.

**Remark B.3.** We now compare our low-rank approximation result with a recent work of (Musco & Woodruff, 2017; Bakshi et al., 2020b). They showed the following theorem:

**Theorem B.1** (Theorem 4.2, (Bakshi et al., 2020b)). *Given a $n \times n$ PSD matrix $A$, target rank $r \in [n]$ and accuracy parameter $\varepsilon \in (0, 1)$, there exists an algorithm that queries $\widetilde{O}(nr/\varepsilon)$ entries in $A$ and with probability at least $9/10$, outputs a rank-$r$ matrix $B$ such that*

$$\|A - B\|_F^2 \le (1 + \varepsilon) \|A - A_r\|_F^2 ,$$

*where $A_r$ is the best rank-$r$ approximation to $A$. Further, the running time is $\widetilde{O}\left(n \left(r/\varepsilon\right)^{\omega-1}\right)$, where $\omega$ is the matrix multiplication constant.*

We note that their result applies to kernel matrices as well via the following fact.

**Fact B.2** (Kernel Matrices are PSD, (Schölkopf et al., 2002)). *Let $k$ be a reproducing kernel and $X$ be $n$ data points in $\mathbb{R}^d$. Let $K$ be the associated $n \times n$ kernel matrix such that $K_{i,j} = k(x_i, x_j)$. Then, $K \succ 0$.*

Here, the family of reproducing kernels is quite broad and includes polynomial kernels, Gaussian, and Laplacian kernel, among others. Therefore, their theorem immediately implies a relative error low-rank approximation algorithm for kernel matrices. Our result and the theorem of (Bakshi et al., 2020b) have comparable runtimes. While (Bakshi et al., 2020b) obtain relative-error guarantees, we only obtain additive-error guarantees.

However, reading each entry of the kernel matrix require $O(d)$ time and thus (Bakshi et al., 2020b) obtain an running time of $\widetilde{O}\left(nd \left(r/\varepsilon\right)^{\omega-1}\right)$, whereas our running time is dominated by $O(nrd/\varepsilon)$. We note that similar ideas as our algorithm for additive error LRA were previously used to design subquadratic algorithms running in time $o(n^2)$ for low-rank approximation of distance matrices (Bakshi & Woodruff, 2018; Indyk et al., 2019).

**Remark B.4.** Our definitions for the local clustering result are adopted from prior literature in property testing; see (Kale & Seshadhri, 2008; Czumaj & Sohler, 2010; Goldreich & Ron, 2011; Czumaj et al., 2015; Chiplunkar et al., 2018; Dey et al., 2019; Peng, 2020; Gluch et al., 2021) and the references within. Our algorithmic details for the local cluster section are also derived from prior works, such as the works of (Czumaj et al., 2015) and (Peng, 2020); indeed, many of the lemmas of the local clustering section follow in a straightforward fashion from (Czumaj et al., 2015) and (Peng, 2020). However, the key difference between these works and our work is that they are in the property testing model where one assumes access to various graph queries in order to design sublinear graph algorithms. To the best of our knowledge, implementation of prior works on local clustering requires having access to the entire neighbor of a vertex when performing random walks, thereby implying the runtime of $\Omega(nd)$ per step of the walk. In contrast, we give efficient constructions of these commonly assumed queries for kernel graphs, rather than assuming oracle access. Indeed, the fact that one can easily take existing algorithms which hold in non kernel settings and apply them to kernel settings in a straightforward manner via our queries can be viewed a major strength of our work.

**Remark B.5.** Our general bound of the number of KDE qeuries required to approximate the total weight of triangles in Theorem E.10 is $\widetilde{O}(m\sqrt{w_G}/w_T)$, where $w_G$ is the sum of all entries of $K$ and $w_T$ is the total weight of triangles we wish to approximate. This bound is a natural generalization of the result of (Eden et al., 2017). There, the goal is to approximate the total number of triangles in an unweighted graph given access to queries of an underlying graph in the form of random vertices and random neighbors of a given vertex (assuming the entire graph is stored in memory). While their model differs from our work, we note that KDE queries constructed in Section C play a similar role to the queries used in (Eden et al., 2017). There the authors give a bound of $\widetilde{O}(m^{3/2}/T)$ queries where $T$ is the total number of triangles. In our case, we indeed get a bound of the order of $m^{3/2}$ in the numerator as $w_G \le m w_{\max}$ and $w_T$ is the natural analogue of $T$ in (Eden et al., 2017). Finally note that under our parameterization of every edge in the kernel graph possessing weight at most $1$ and at least $\tau$, our bound reduces to at most $\widetilde{O}(1/\tau^3)$ KDE queries.

We finally note that to the best of our knowledge, all prior works for approximating the number of triangles in a graph require the full graph to be instantiated, which implies a lower bound of time $\Omega(n^2 d)$ in our setting.

We also note that our paper is closely related to the field of (graph) property testing. In graph property testing, it is customary to assume query access to an unknown graph via vertex and edge queries (Goldreich, 2017). While specific details vary, common queries include access to random vertices and random neighbors of a given vertex, among others. The goal of the field is to design algorithms that require queries sublinear in $n$, the number of vertices, or $n^2$, the size of the graph. We can interpret the graph primitives we construct as a realization of the property testing model where queries are explicitly constructed.

## B.1  Preliminaries

First, we discuss the cost of constructing KDE data structure and performing the queries described in Definition 1.1. Table 1 summarizes previous work on kernel density estimation though for the sake of uniformity, we list only "high-dimensional" data structures, whose running times are polynomial in the dimension $d$. Those data structures have construction times of the form $O(dn/(\tau^p \varepsilon^2))$ and answer KDE queries in time $O(d/(\tau^p \varepsilon^2))$, under the condition that for all queries $y$ we have $\frac{1}{n} \sum_{x \in X} k(x, y) \geq \tau$ (which clearly holds under our Parameterization 1.1). The algorithms are randomized, and report correct answers with a constant probability. The values of $p$ lie in the interval $[0, 1)$, and depend on the kernel. For comparison, note that a simple random sampling approach, which selects a random subset $R \subset X$ of size $O(1/(\tau \varepsilon^2))$ and reports $\frac{n}{|R|} \sum_{x \in R} k(x, y)$, achieves the exponent of $p = 1$ for any kernel whose values lie in $[0, 1]$.

We view our algorithms as parameterized in terms of $\tau$, the smallest edge length. We argue this is a natural parameterization. When picking a kernel function $k$, we also have to pick a *scale* term $\sigma$ (for example, the exponential kernel is of the form $k(x, y) = \exp(-\|x - y\|_2/\sigma)$). In practice, a common choice of $\sigma$ follows the so called 'median rule' where $\sigma$ is set to be the median distance among all pairs of points in $X$. Thus, according to the median rule, the 'typical' kernel values in the graph $K$ are $\Omega(1)$. While this is only true for 'typical,' and not all, edge weights in $K$, we believe the KDE query abstraction of Definition 1.1 still provides nontrivial and useful algorithms for working with kernel graphs. Typically in practice, the setting of $\tau$ is a small constant, independent of the size of the dataset (Karppa et al., 2022).

We note that, in addition to the aforementioned algorithms with theoretical guarantees, there are other practical algorithms based on random sampling, space partition trees (Gray & Moore, 2001; 2003; Lee et al., 2006; Lee & Gray, 2008; Morariu et al., 2008; Ram et al., 2009; March et al., 2015), coresets (Phillips, 2013; Zheng et al., 2013; Phillips & Tai, 2020b), or combinations of these methods (Karppa et al., 2022), which support queries needed in Definition 1.1; see (Karppa et al., 2022) for an in-depth discussion on applied works.

While these algorithms do not necessarily have as strong theoretical guarantees as the ones discussed above and in Table 1, we can nonetheless use them via black box access in our algorithms and utilize their practical benefits.

## C  Algorithmic Building Blocks

## C.1  Multi-level KDE

We first describe the "multi-level" KDE data structure, which is required in our algorithms. The data structure recursively constructs a KDE data structure on the entire dataset $X$, and then recursively partitions $X$ into two halves, building a KDE data structure on each half. See Algorithm 1 for more details.

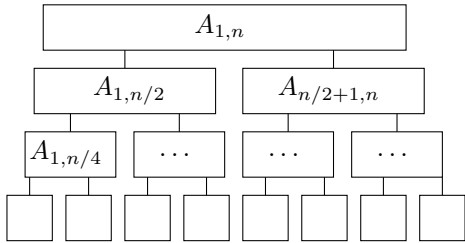

Figure 1: Multi-level Kernel Density Estimation Data Structure.

---

**Algorithm 1** Multi-level KDE Construction

---

**Require:** Input dataset $X \subset \mathbb{R}^d$, precision $\varepsilon > 0$
1: $T = X$
2: **while** $|T| > 1$ **do**
3:     Construct $\text{KDE}_X$ queries
4:     Recursively apply Multi-level KDE Construction to $T[1 : \lfloor m/2 \rfloor]$ and $T[\lfloor m/2 \rfloor + 1 : m]$
5: **end while**
6: **Return** all the data structures associated with the KDE query constructions

---

**Lemma C.1.** *Given a dataset $X \subset \mathbb{R}^d$, suppose the initialization of the KDE data structure defined in Definition 1.1 uses runtime $f(n, \varepsilon)$ for some function linear in $n$. Then the total construction time of Algorithm 1 is $f(n \log n, \varepsilon)$.*

*Proof.* The proof follows from the fact that at each recursive level, we do $O(f(n, \varepsilon))$ total work since $f$ is linear in $n$ and there are $O(\log n)$ levels. $\square$

## C.2   Weighted Vertex Sampling

We now discuss our fundamental primitives. The first one is sampling vertices by their (weighted) degree.

**Definition C.1** (Weighted Vertex Sampling). *The weighted degree of a vertex $x_i$ with $i \in [n]$ is $w_i = \sum_{j \neq i} k(x_i, x_j)$. The goal of weighted vertex sampling is to output a vertex $v$ such that $\Pr[v = x_i] = \frac{(1 \pm \varepsilon) w_i}{\sum_{j \in [n]} w_j}$ for all $i \in [n]$.*

This is a straightforward application of using $n$ KDE queries to get the (weighted) vertex degree of all $n$ vertices. Note that this only takes $n$ queries and only has to be *done once*. Therefore, we can think of vertex sampling as a preprocessing step that uses $O(n)$ queries upfront and then allows for arbitrary access at any point in the future with no query cost.

---

**Algorithm 2** Vertex Sampling by (Weighted) Degree

---

**Require:** Precision $\varepsilon$
**Ensure:** Reals $p_i$ such that $(1 - \varepsilon)\deg(x_i) \leq p_i \leq (1 + \varepsilon)\deg(x_i)$ for all $1 \leq i \leq n$
1: **for** $i = 1$ to $i = n$ **do**
2:     $p_i \leftarrow \text{KDE}_X(x_i) - (1 - \varepsilon) \, k(x_i, x_i)$.
3: **end for**
4: **Return** $\{p_i\}_{i=1}^n$

---

Once we acquire $\{p_i\}_{i=1}^n$, we can perform a fast sampling procedure through the following algorithm, which we state in slightly more general terms.

---

**Algorithm 3** Sample from Positive Array

---

**Require:** Input array $A = [a_1, \cdots, a_n]$ with $a_i > 0$ for all $i$. Access to queries $A_{i,j} = \sum_{i \leq t \leq j} a_t$
     for $1 \leq i \leq j \leq n$.
 1: $T = A$
 2: **while** $|T| > 1$ **do**
 3:     $m = \text{len}(T)$
 4:     $a \leftarrow \sum(T[1 : \lfloor m/2 \rfloor])$ // Can be simulated using an $A_{i,j}$ query
 5:     $b \leftarrow \sum(T[\lfloor m/2 \rfloor + 1 : m])$
 6:     **if** $\text{Unif}[0, 1] \leq a/(a + b)$ **then**
 7:        $T \leftarrow T[1 : \lfloor m/2 \rfloor]$
 8:     **else**
 9:        $T \leftarrow T[\lfloor m/2 \rfloor + 1 : m]$
10:     **end if**
11: **end while**
12: **Return** the single element in $T$

---

Combining Algorithms 2 and 3, we can perfectly sample from the degree distribution of the graph $K$.

---

**Algorithm 4** Faster Degree Sampling

---

**Require:** Reals $p_i$ such that $(1 - \varepsilon)\deg(x_i) \leq p_i \leq (1 + \varepsilon)\deg(x_i)$ for all $1 \leq i \leq n$
 1: $i \leftarrow$ index in $[n]$, which is the output of running Algorithm 3 on the array $\{p_i\}_{i=1}^n$
 2: **Return** $x_i$

---

We now analyze the correctness and the runtimes of the algorithms proposed in Section C. First, we give guarantees on Algorithm 2.

**Theorem C.2.** *Algorithm 2 returns $\{p_i\}_{i=1}^n$ such that $(1 - \varepsilon)\deg(x_i) \leq p_i \leq (1 + \varepsilon)\deg(x_i)$ for all $1 \leq i \leq n$.*

*Proof.* The proof follows by the Definition of a KDE query, Definition 1.1. $\qquad\square$

We now analyze Algorithm 3, which samples from an array based on a tree data structure given access to consecutive sum queries. The analysis of this process will also greatly facilitate the analysis of other algorithms from Section C.

**Lemma C.3.** *Algorithm 3 samples an index $i \in [n]$ proportional to $a_i$ in $O(\log n)$ time with $O(\log n)$ queries.*

*Proof.* Consider the sampling diagram given in Figure 1. Algorithm 3 does the following: it first queries the root node $A_{1,n}$ and then its two children $A_{1,m}, A_{m+1,n}$ where $m = \lfloor n/2 \rfloor$. Note that $A_{1,n} = A_{1,m} + A_{m+1,n}$. It then picks the tree rooted at $A_{1,m}$ with probability $\frac{\sum_{i \in [m]} a_i}{\sum_{i \in [n]} a_i}$ and otherwise, picks the tree rooted at $A_{m+1,n}$. The procedure recursively continues by querying the root node, its two children, and picking one of its children to be the new root node with probability proportional to the child's weight given by an appropriate query access. This is done until we reach a leaf node that corresponds to an index $i \in [n]$.

We now prove correctness. Note that each node of the tree in Figure 1 corresponds to a subset $S \subseteq [n]$. We prove inductively that the probability of landing on the vertex is equal to $\sum_{i \in S} a_i$. This is true for the root node of the tree since the algorithm begins at the root note. Now consider transitioning from some node $S$ to one of its children $S_1, S_2$. We know that we are at node $S$ with probability $\sum_{i \in S} a_i / \sum_j a_j$. Furthermore, we transition to $S_1$ with probability $\sum_{i \in S_1} a_i / \sum_{j \in S} a_j$. Therefore, the probability of being at $S_1$ is equal to

$$\frac{\sum_{i \in S_1} a_i}{\sum_{j \in S} a_j} \cdot \frac{\sum_{i \in S} a_i}{\sum_j a_j} = \frac{\sum_{i \in S_1} a_i}{\sum_j a_j}.$$

Since there is only one path from the root node to any vertex of a tree, this completes the induction.

The runtime and the number of queries taken follows from the fact that the sampling procedure descends on a tree with $O(\log n)$ height. $\qquad\square$

Combining Algorithms 2 and 3 allows us to sample from the degree distribution of the graph $K$ up to low error in total variation (TV) distance.

**Theorem C.4.** *Algorithm 4 samples from the degree distribution of $K$ up to TV error $O(\varepsilon)$ using a fixed overhead of $n$ KDE queries and runtime $O(\log n)$.*

*Proof.* Since $p_i$ is with a $1 \pm \varepsilon$ factor of $\deg(x_i)$ for all $i$, then $\{p_i\}_{i=1}^n$ is $O(\varepsilon)$ close in total variation distance from the true degree distribution. Moreover, Algorithm 3 perfectly samples from the array $\{p_i\}_{i=1}^n$, which proves the first part of the theorem.

For the second part, note that acquiring $\{p_i\}_{i=1}^n$ requires $n$ KDE queries. We can then construct the data structure for Algorithm 3 by computing all the partial prefix sums in $O(n)$ time. Now the query access required by Algorithm 3 can be computed in $O(1)$ time through an appropriate subtraction of two prefix sums. Note that the previous steps need to be only done *once* and can be utilized for all future runs of Algorithm 3. It follows from Lemma C.3 that Algorithm 4 takes $O(\log n)$ time. $\qquad\square$

## C.3   Weighted Edge Sampling and Weighted Neighbor Edge Sampling

We describe how to perform weighted neighbor edge sampling.

**Definition C.2** (Weighted Neighbor Edge Sampling). *Given a vertex $x_i$, the goal of weighted neighbor edge sampling is to output a vertex $v$ such that $\Pr[v = x_k] = \frac{(1 \pm \varepsilon) k(x_i, x_k)}{\sum_{j \in n, j \neq i} k(x_i, x_j)}$ for all $i \in [n]$.*

---

**Algorithm 5** Sample Random Neighbor

---

**Require:** Input vertex $x_i \in X$, precision $\varepsilon$
**Ensure:** $x \in X \setminus \{x_i\}$ such that the probability of selecting $x$ is proportional to $k(x_i, x)$
 1: $\varepsilon' = \varepsilon / \log n$
 2: **while** $|T| > 1$ **do**
 3: $\quad m \leftarrow |T|$
 4: $\quad a \leftarrow \text{KDE}_{T[1:m/2], \varepsilon'}(x_i)$
 5: $\quad b \leftarrow \text{KDE}_{T[m/2+1:m], \varepsilon'}(x_i)$
 6: $\quad$ **if** $x_i \in T[1 : m/2]$ **then**
 7: $\quad\quad a \leftarrow a - (1 - \varepsilon') k(x_i, x_i)$
 8: $\quad$ **end if**
 9: $\quad$ **if** $x_i \in T[m/2 + 1 : m]$ **then**
10: $\quad\quad b \leftarrow b - (1 - \varepsilon') k(x_i, x_i)$
11: $\quad$ **end if**
12: $\quad$ **if** $\text{Unif}[0, 1] \leq a/(a + b)$ **then**
13: $\quad\quad T \leftarrow T[1 : m/2]$
14: $\quad$ **else**
15: $\quad\quad T \leftarrow T[m/2 + 1 : m]$
16: $\quad$ **end if**
17: **end while**
18: **Return** the single element in $T$

---

We now prove the correctness of Algorithm 5 based on the ideas in Lemma C.3. Note that Algorithm 5 takes in input a precision level $\varepsilon$, which can be adjusted and impacts the accuracy of KDE queries. We will discuss the cost of initializing KDE queries with various precisions in Section B.1.

**Theorem C.5.** *Let $x_i \in X$ be an input vertex. Consider the distribution $\mathcal{D}$ over $X \setminus \{x_i\}$, the neighbors of $x_i$ in the graph $K$, induced by the edge weights in $K$. Algorithm 5 samples a neighbor from a distribution that is within TV distance $O(\varepsilon)$ from $\mathcal{D}$ using $O(\log n)$ KDE queries and $O(\log n)$ time. In addition, we can perfectly sample from $\mathcal{D}$ using $O(\log n/\tau)$ additional kernel evaluations in expectation.*

*Proof.* The proof idea is similar to that of Lemma C.3. Given a vertex $x_i$, its adjacent edges have associated weights and our goal is to sample an edge proportion to these weights. However, unlike the degree case, performing edge sampling is not a straightforward KDE query as an edge only cares about the kernel value between two points, rather than the sum of kernel values that a KDE query provides. Nevertheless, we can utilize the tree procedure outline in the proof of Lemma C.3 in conjunction with KDE queries with over various subsets of $X$.

Imagine the same tree as in Figure 1 where each subset corresponds to a subset of neighbors of $x_i$ (note that $x_i$ cannot be its own neighbor and hence we subtract $k(x_i, x_i)$ in line 7 or line 10). Algorithm 5 descends down the tree using the same probabilistic procedure as in the proof of Lemma C.3: at every node, it picks one of the children to descend to with probability proportional to its weight. Here, the weight of a child node in the tree in Figure 1 is the sum of the weights of the edges connecting to the corresponding neighbors of $x_i$.

Now compare the telescoping product of probabilities that lands us in some leaf node $a_j$ to the ideal telescoping product if we knew the exact array of edge weights as in the proof of Lemma 6. Suppose the tree has height $\ell$. At each node in our actual path descending down the tree, we take the next step according to the ideal descent (according to the ideal telescoping product), with the same probability, except for possibly an overestimate or underestimate by a factor of $1 + \varepsilon'$ or $1 - \varepsilon'$ factor respectively.

Therefore, we land in the correct leaf node with the same probability as in the ideal telescoping product, except our probability can be off by a multiplicative $(1 \pm \varepsilon')^\ell$ factor. However, since $\varepsilon' = \varepsilon / \log n$ and $\ell \leq \log n$, this factor is within $1 \pm \varepsilon$. Thus, we sample from the correct distribution over the leaves of the trees in Figure 1 up to TV distance $O(\varepsilon)$. Now by doing $O(1/\tau)$ steps of rejection sampling, we can actually get a prefect sample of the edge. This is because the denominator of the fraction for $\Pr[v = x_k]$ is at least $\Omega(n\tau)$ and at most $n$ so we can estimate the proportionality constant in the denominator by $n$ which is only at most $O(1/\tau)$ multiplicative factor larger. Hence by standard guarantees of rejection sampling, we only need repeat the sampling procedure $O(1/\tau)$ additional times. $\qquad\square$

---

**Algorithm 6** Sample Random Edge by Weight

---

**Ensure:** Sample edge $(x_i, x_j)$ with probability at least $(1 - \varepsilon)k(x_i, x_j)$
1: $v \leftarrow$ random vertex by using Algorithm 4
2: $w \leftarrow$ random Neighbor of $v$ using Algorithm 5.
3: **Return:** Edge $(v, w)$.

---

**Theorem C.6** (Weighted Edge Sampling). *Algorithm 6 returns a random edge of $K$ with probability proportional to at least $(1 - \varepsilon)$ its weight using 1 call to Algorithm 5.*

*Proof.* Consider an edge $(u, v)$. The vertex $u$ is sampled with probability at least $(1 - 2\varepsilon)\deg(u)/\sum_{x \in X} \deg(x)$. Given this, the vertex $v$ is then sampled with probability at least $(1 - 2\varepsilon)\frac{k(u,v)}{\sum_{x \in X \setminus u} k(u,x)} = (1 - 2\varepsilon)\frac{k(u,v)}{\deg(u)}$. Using the same analysis for sampling $v$ and then $u$, we have that any edge $(u, v)$ is sampled with probability at least $1 - 2\varepsilon$ times $k(u, v)/\sum_{e \in K} w(e)$. Note that the same rejection sampling remark as in the proof of Theorem C.5 applies and we can perfectly sample an edge proportional to its weight with an addition $O(1/\tau)$ rejection sampling steps. $\qquad\square$

## C.4   Random Walk

**Theorem C.7.** *Algorithm 7 outputs a vertex from a vertex within $O(T\varepsilon)$ total variation distance from the true random walk distribution. Each step of the walk requires 1 call to Algorithm 5 .*

*Proof.* The proof follows from the correctness of Algorithm 5 given in Theorem C.5. Lastly we again note that by performing an additional $O(1/\tau)$ rounds of rejection sampling steps (as outlined in the end of the proof of Theorem C.5), we can make sure that we are sampling from the *true* random walk distribution at each step of the walk. $\qquad\square$

---

**Algorithm 7** Perform Random Walk

---

**Require:** Input vertex $x_i \in X$, length of walk $T \geq 1$.
 1: Start at vertex $x_i$
 2: $v \leftarrow x_i$
 3: **for** $j = 1$ to $T$ **do**
 4:    $w \leftarrow$ Sample random neighbor of $v$ (Algorithm 5)
 5:    $v \leftarrow w$
 6: **end for**
 7: **Return** $v$

---

# D   Linear Algebra Applications

We now present a wide array of applications of the algorithmic building blocks constructed in Section C. Altogether, these applications allow us to understand or approximate fundamental and properties of the kernel matrix and the graph $K$. In this section we present the linear algebra applications and the graph applications are given in Section E.

## D.1   Spectral Sparsification

---

**Algorithm 8** Spectral Sparsification of the Kernel Graph

---

 1: Let $t = O(n \log(n)/\varepsilon^2 \tau^3)$ be the number of edges that are to be sampled
 2: Let $\hat{p}$ denote the distribution returned by Algorithm 2 for a small enough constant $\varepsilon$
 3: Initialize $G' = \emptyset$
 4: **for** $i = 1, \ldots, t$ **do**
 5:    Sample a vertex $u$ from the distribution $\hat{p}$
 6:    Sample a neighbor $v$ of $u$ using Algorithm 5 with constant $\varepsilon$
 7:    Let $\hat{q}_{uv}$ be the probability that Algorithm 5 samples $v$ given $u$ as input
 8:    Similarly define and compute $\hat{q}_{vu}$
 9:    $w_{uv} = 1/(t(\hat{p}_u \hat{q}_{uv} + \hat{p}_v \hat{q}_{vu}))$
 10:    Add the weighted edge $(\{u, v\}, w_{uv})$ to the graph $G'$
 11: **end for**
 12: Compute an $\varepsilon/2$ spectral sparsifier $G''$ of graph $G'$ using (Batson et al., 2013)
 13: **return** $G''$

---

Given a set $X$, $|X| = n$, and a kernel $k : X \times X \to \mathbb{R}^+$, we describe how to construct a spectral sparsifier for the weighted complete graph on $X$ where weight of the edge $\{x_i, x_j\}$ is given by $k(x_i, x_j)$.

**Definition D.1** (Graph Laplacian). *Given a weighted graph $G = (V, E, w)$, the Laplacian of $G$, denoted by $L_G = D - A$, where $A$ is the adjacency matrix of $G$ with $A_{i,j} = w(\{i, j\})$ and $D$ is a diagonal matrix such that for all $i \in [n]$, $D_{i,i} = \sum_{j \neq i} A_{i,j}$.*

**Theorem D.1** (Spectral Sparsification of Kernel Density Graphs). *Given a dataset $X$ of $n$ points in $\mathbb{R}^d$, and a kernel $k : X \times X \to \mathbb{R}^+$, let $G = (X, \binom{X}{2}, w)$ be the weighted complete graph on $X$ with the weights $w(\{x_i, x_j\}) = k(x_i, x_j)$. Further, for all $x_i, x_j \in X$, let $k(x_i, x_j) \geq \tau$, for some $\tau \in (0, 1)$. Let $L_G$ be the Laplacian matrix corresponding to the graph $G$. Then, for any $\varepsilon \in (0, 1)$, Algorithm 8 outputs a graph $G''$ with only $m = O(n \log n/(\varepsilon^2 \tau^3))$ edges, such that with probability at least $9/10$,*

$$(1 - \varepsilon)L_G \preceq L_{G'} \preceq (1 + \varepsilon)L_G.$$

*The algorithm makes $\widetilde{O}(m/\tau^3)$ KDE queries and requires $\tilde{O}(md/\tau^3)$ post-processing time.*

Let $G_d$ be the weighted directed graph obtained by arbitrarily orienting the edges of the graph $G$ and let $H$ be an edge-vertex incidence matrix defined as follows : for each $e = (x_i, x_j)$ in graph $G_d$, let $H_{e,x_i} = \sqrt{k(x_i, x_j)}$ and $H_{e,x_j} = -\sqrt{k(x_i, x_j)}$. Note that $H^\top H = L_G$. Our idea to construct

spectral sparsifier is to compute a sampling-and-reweighting matrix $S$, i.e., a matrix that has at most one nonzero entry in each row, that with probability $\geq 9/10$, satisfies

$$(1-\varepsilon)L_G = (1-\varepsilon)H^\top H \preceq H^\top S^\top SH \preceq (1+\varepsilon)H^\top H = (1+\varepsilon)L_G.$$

The edges sampled by $S$ form the edges of the graph $G'$. We construct this matrix $S$ by sampling rows of the matrix $H$ from a distribution close to the distribution that samples a row of $H$ with a probability proportional to its squared norm. We show that this gives a spectral sparsifier by showing that such a distribution approximates the "leverage score sampling" distribution.

**Definition D.2** (Leverage Scores). *Let $M$ be a $n \times d$ matrix and $m_i$ denote the $i$-th row of $M$. Then, for all $i \in [n]$, $\tau_i$, the $i$-th leverage of $M$ is defined as follows:*

$$\tau_i = m_i(M^\top M)^+ m_i^\top,$$

*where $X^+$ is the Moore-Penrose pseudoinverse for a matrix $X$.*

We introduce the following intermediate lemmas. We begin by recalling that sampling edges proportional to leverage scores (effective resistances on a graph) suffices to obtain spectral sparsification (Spielman & Srivastava, 2011; Woodruff, 2014).

**Lemma D.2** (Leverage Score Sampling implies Sparsification). *Given an $n \times d$ matrix $M$ and $\varepsilon \in (0,1)$, for all $i \in [t]$, let $\tau_i$ be the $i$-th leverage score of $M$. Let $p = \{p_1, p_2, \ldots, p_n\}$ be a distribution over the rows of $M$ such that $p_i = \tau_i / \sum_{j \in [n]} \tau_j$. Further, for some $\phi \in (0,1)$, let $\hat{p} = \{\hat{p}_1, \hat{p}_2, \ldots, \hat{p}_n\}$ be a distribution such that $\hat{p}_i \geq \phi p_i$ and let $t = O\left(\frac{d \log(d)}{\varepsilon^2 \phi}\right)$. Let $S \in \mathbb{R}^{t \times n}$ be a random matrix where for all $j \in [t]$, the $j$-th row is independently chosen as $(1/\sqrt{t\hat{p}_i})e_i^\top$ with probability $\hat{p}_i$. Then, with probability at least $99/100$,*

$$(1-\varepsilon)M^\top M \preceq M^\top S^\top SM \preceq (1+\varepsilon)M^\top M.$$

Next, we show that the matrix $H$ is well-conditioned, in fact the condition number is independent of the dimension and only depends on the minimum kernel value between any two points in the dataset. This lets us use our edge sampling routines to compute an $\varepsilon$ spectral sparsifier.

**Lemma D.3** (Bounding Condition Number). *Let $H$ be the edge-vertex incidence matrix as defined and also has the property that all nonzero entries in the matrix have an absolute value of at most $1$ and at least $\sqrt{\tau}$. Let $\sigma_{\max}(H)$ be the maximum singular value of $H$ and $\sigma_{\min}(H)$ be the minimum nonzero singular value of $H$. Then $\sigma_{\max}(H)/\sigma_{\min}(H) \leq 4\sqrt{2}/\tau^{1.5}$.*

*Proof.* We use the following standard upper bound on the spectral norm of an arbitrary matrix $A$ to upper bound the spectral norm of the matrix $H$:

$$\|A\|_2 \leq \sqrt{\left(\max_i \sum_j |A_{i,j}|\right)\left(\max_j \sum_i |A_{i,j}|\right)}.$$

For the matrix $H$, as each column has at most $n$ nonzero entries and each row has at most 2 nonzero entries and from the assumption that all the entries have magnitude at most 1, we obtain that $\|H\|_2 \leq \sqrt{2n}$. To obtain lower bounds on $\sigma_{\min}(H)$, we appeal to a Cheeger-type inequality for weighted graphs from (Friedland, 1992; Friedland & Nabben, 2002). First, we note that $\sigma_{\min}(H) = \sqrt{\sigma_{\min}(H^\top H)} = \sqrt{\sigma_{\min}(L_G)}$ where $G$ is the kernel graph that we are considering with each edge having a weight of at least $\tau$. Let $0 = \lambda_1 \leq \lambda_2 \leq \cdots \leq \lambda_n$ be the eigenvalues of the positive semi-definite matrix $L_G$. Now we have that

$$\sigma_{\min}(L_G) = \lambda_2(L_G) \geq \min_i(\delta_i/2)\varepsilon(G)^2$$

where $\delta_i = \sum_{j \neq i} k(x_i, x_j)$ i.e., the weighted degree of vertex $x_i$ in graph $G$ and

$$\varepsilon(G) = \min_{\phi \neq U \subset V, |U| \leq n/2} \frac{|E(U, \bar{U})|}{|E(U)|}$$

where $|E(U)|$ denotes the sum of weighted degrees of vertices in $U$ and $|E(U, \bar{U})|$ denotes the total weight of edges with one end point in $U$ and the other outside $U$. Using the fact that $G$ is a

complete graph with each edge having a weight of at least $\tau$ and at most 1, we obtain $|E(U, \bar{U})| \geq \tau|U||\bar{U}|$ and $|E(U)| \leq n|U|$, which implies that $\varepsilon(G) \geq \min_{\phi \neq U \subset V, |U| \leq n/2} \tau|\bar{U}|/n \geq \tau/2$. We also similarly have that $\min_i \delta_i \geq (n-1)\tau$, which overall implies that $\lambda_2(L_G) \geq n\tau^3/16$ and that $\sigma_{\min}(H) \geq \sqrt{n}\tau^{1.5}/4$. Thus, we obtain that $\sigma_{\max}(H)/\sigma_{\min}(H) \leq 4\sqrt{2}/\tau^{1.5}$. $\qquad\square$

We are now ready to complete the proof of our main theorem:

*Proof of Theorem D.1.* Let $q = \{q_1, q_2, \ldots, q_{\binom{n}{2}}\}$ be a distribution over the rows of $H$ such that for all edges $e = \{i, j\}$, $q_e \geq c\frac{\|H_{e,*}\|_2^2}{\|H\|_F^2} = \frac{k(x_i, x_j)}{\sum_{e' = \{i', j'\}} k(x_{i'}, x_{j'})}$, for a fixed universal constant $c$.

Next, we show that this distribution is $\Theta(1/\kappa^2)$ approximation to the leverage score distribution for $H$. Let $H = U\Sigma V^\top$ be the "thin" singular value decomposition of $H$ and therefore all the diagonal entries of $\Sigma$ are nonzero. By definition $\tau_i = \|U_{i*}\|_2^2$. We have

$$\|h_i\|_2^2 = \|U_{i*}\Sigma V^\top\|_2^2 = \|U_{i*}\Sigma\|_2^2$$

where the equality follows from the fact that $V^\top$ has orthonormal rows. Now, $\|U_{i*}\Sigma\|_2^2 \geq \|U_{i*}\|_2^2\sigma_{\min}^2$ and $\|U_{i*}\Sigma\|_2^2 \leq \|U_{i*}\|_2^2\sigma_{\max}^2$. Therefore, for all $i \in \binom{n}{2}$, defining $\kappa = \sigma_{\min}/\sigma_{\max}$, we have

$$\frac{\tau_i}{\sum_j \tau_j} = \frac{\|U_{i*}\|_2^2}{\sum_j \|U_{j*}\|_2^2} \geq \frac{\|h_i\|_2^2/\sigma_{\max}^2}{\sum_j \|h_j\|_2^2/\sigma_{\min}^2} = \frac{1}{\kappa^2}\frac{\|h_i\|_2^2}{\|H\|_F^2}.$$

Then, we invoke Lemma D.2 with $\phi = \Omega(1/\kappa^2)$ and conclude that sampling $t = O\left(\frac{n \log n}{\varepsilon^2 \kappa^2}\right)$ rows of $H$ results in a sparse graph $G'$ with corresponding Laplacian $L_{G'}$ such that with probability at least $99/100$,

$$(1 - \varepsilon/2)L_G \preceq L_{G'} \preceq (1 + \varepsilon/2)L_G.$$

Further, by Lemma D.3, we can conclude $\kappa^2 \leq 32/\tau^3$ and thus sampling $t = O\left(\frac{n \log n}{\varepsilon^2 \tau^3}\right)$ edges suffices.

We do not use Algorithm 6 to sample random edges from the perfect distribution to implement spectral sparsification as we cannot compute the exact sampling probability of the edge that is sampled. So, we first use Algorithm 4 with constant $\varepsilon$ (say 1/2) to sample a vertex $u$ and Algorithm 5 with constant $\varepsilon$ (say 1/2) to sample a neighbor $v$ of $u$. Note that Algorithms 4 and Algorithms 5 can be modified to also return the probabilities $\hat{p}_u$ and $\hat{q}_{vu}$ with which the vertex $i$ and the neighbor $j$ of $i$ are sampled. We can further query the algorithms to return $\hat{p}_v$ and $\hat{q}_{uv}$. Now, $q_{\{u,v\}} = \hat{p}_u\hat{q}_{vu} + \hat{p}_v\hat{q}_{uv}$ is the probability with which this sampling process samples the edge $\{u, v\}$ and we have that $\hat{p}_u\hat{q}_{vu} + \hat{p}_v\hat{q}_{uv} \geq c\frac{k(x_u, x_v)}{\sum_{i \neq j} k(x_i, x_j)}$ and we use this distribution $q$ to implement spectral sparsification as described above. As already seen (Theorem C.5), to compute vertex sampling distribution $\hat{p}$, we use $n$ KDE queries and for each neighbor sampling step, we use $O(\log n)$ KDE queries. Thus, we overall use $O(n \log^2 n/(\varepsilon^2\tau^3))$ constant approximate KDE queries to obtain an $\varepsilon$ spectral sparsifier. $\qquad\square$

We can further compute another graph $G''$ with only $O(n/\varepsilon^2)$ edges by computing an $\varepsilon/2$ spectral sparsifier for $G'$ using the deterministic spectral sparsification algorithm of (Batson et al., 2013). Conditioning on the graph $G'$ being an $\varepsilon$ spectral sparsifier of $G$, the graph $G''$ will then be an $\varepsilon$ sparsifier for $G$. This procedure doesn't require any KDE queries and solely operates on the weighted graph $G'$.

**Hardness for spectral sparsification.** We observe that we can use the lower bound from Alman et. al. to establish hardness in terms of $\tau$ from Parameterization 1.1. The lower bound we obtain is as follows:

**Theorem D.4** (Lower Bound for Spectral Sparsification under Parameterization 1.1). *Let $k$ be the Gaussian kernel and let $X$ be dataset such that $\min_{x,y \in X} k(x, y) = \tau$, for some $1 > \tau > 0$. Then, any algorithm that with probability $9/10$ outputs an $O(1)$-approximate spectral sparsifier for the*

*kernel graph associated with $X$, with $O(n^{2-\delta})$ edges, where $\delta < 0.01$ is a fixed universal constant, requires $\Omega\left(n \cdot 2^{\log(1/\tau)^{0.32}}\right)$ time, assuming the strong exponential time hypothesis.*

First, we begin with the definition of a multiplicatively-Lipschitz function:

**Definition D.3** (Multiplicatively-Lipschitz Kernels). *A kernel $k$ over a set $X$ is $(c, L)$-multiplicatively Lipschitz if for any $\rho \in (1/c, c)$, and for any $x, y \in X$, $c^{-L}k(x, y) \leq k(\rho x, \rho y) \leq c^L k(x, y)$.*

We will require the following theorem showing hardness for sparsification when the kernel function is not multiplicatively-Lipschitz:

**Theorem D.5** (Theorem 8.3 (Alman et al., 2020)). *Let $k$ be a function and $X$ be a dataset such that $k$ is not $(c, L)$-multiplicatively-Lipschitz on $X$ for some $L > 1$ and $c = 1 + 2\log\left(10 \cdot 2^{L^{0.48}}\right)/L$. Then, there is no algorithm that returns a sparsifier of the kernel graph associated with $X$ with $O(n^{2-\delta})$ edges, where $\delta < 0.01$ is a fixed universal constant, in less than $O\left(n \cdot 2^{L^{0.48}}\right)$ time, assuming the strong exponential time hypothesis.*

*Proof of Theorem D.4 .* First, we show that for any $c > 1$, if $L < \log(1/\tau)(c-1)$, then the Gaussian kernel $k$ is not $(c, L)$-multiplicatively Lipschitz. Let $z = \|x - y\|_2^2$ and let $f(z) = e^{-z}$. Observe, it suffices to show that there exists a $z$ such that $f(cz) \leq c^{-L}f(z)$. Let $z$ be such that $f(z) = e^z = \min_{x,y} k(x, y) = \tau$, i.e. $z = \log(1/\tau)$. Then,

$$f(c\log(1/\tau)) = e^{-c\log(1/\tau)},$$

and for $L < \log(1/\tau)(c - 1)$

$$c^{-L}f(\log(1/\tau)) > e^{-c\log(1/\tau)}.$$

Then, applying Theorem D.5 with $c = 1 + \frac{1}{\sqrt{L}}$, it suffices to conclude $k$ is not $(c, L)$-multiplicatively Lipschitz when $L < \log^{2/3}(1/\tau)$, which concludes the proof. $\square$

### D.1.1 Solving Laplacian Systems Approximately

We describe how to approximately solve the Laplacian system $L_G x = b$ using the spectral sparsifier $L_{G'}$. First, we note the following theorem that states the running time and approximation guarantees of fast Laplacian solvers.

**Theorem D.6** ((Koutis et al., 2011), (Spielman & Teng, 2004)). *There is an algorithm that takes an input a graph Laplacian $L$ of a graph with $m$ weighted edges, a vector $b$, and an error parameter $\alpha$ and returns $x$ such that with probability at least $99/100$,*

$$\|x - L^+ b\|_L \leq \alpha \|L^+ b\|_L,$$

*where $\|x\|_L = \sqrt{x^\top L x}$. The algorithm runs in time $\widetilde{O}(m \log(1/\alpha))$.*

We have the following theorem that bounds the difference between solutions for the exact Laplacian system and the spectral sparsifier Laplacian.

**Theorem D.7.** *Let $L_G$ be the Laplacian of a connected graph $G$ on $n$ vertices and let $L_{G'}$ be the Laplacian of an $\varepsilon$-spectral sparsifier $G'$ of graph $G$ i.e.,*

$$(1 - \varepsilon)L_G \preceq L_{G'} \preceq (1 + \varepsilon)L_G,$$

*for $\varepsilon < c$ for a small enough constant $c$. Then, for any vector $b$ with $\mathbf{1}^\top b = 0$, $\|L_G^+ b - L_{G'}^+ b\|_{L_G} \leq 2\sqrt{\varepsilon}\|L_G^+ b\|_{L_G}$.*

*Proof.* Note that for $\varepsilon < 1$, the graph $G'$ also has to be connected and therefore the only eigen vectors corresponding to eigen value 0 of the matrices $L_G$ and $L_{G'}$ are of the form $a \cdot \mathbf{1}$ for $a \neq 0$

and hence columns (and rows) of $L_G$ span all vectors orthogonal to $1$. Therefore $L_G L_G^+ = L_G^+ L_G = I - (1/n)11^\top$. Now,

$$
\begin{aligned}
\|L_G^+ b - L_{G'}^+ b\|_{L_G}^2 &= b^\top (L_G^+ - L_{G'}^+) L_G (L_G^+ - L_{G'}^+) b \\
&= b^\top L_G^+ L_G L_G^+ b - b^\top L_{G'}^+ L_G L_G^+ b - b^\top L_{G'}^+ L_G L_G^+ b + b^\top L_{G'}^+ L_G L_{G'}^+ b \\
&\leq b^\top L_G^+ b - b^\top L_{G'}^+ b - b^\top L_{G'}^+ b + \frac{1}{1-\varepsilon} b^\top L_{G'}^+ b
\end{aligned}
$$

where in the last inequality, we used $L_G L_G^+ b = 1$ and that for any vector $v$, $v^\top L_G v \leq \frac{1}{1-\varepsilon} v^\top L_{G'} v$. As the null spaces of both $L_G$ and $L_{G'}$ are given by $\{a1 \mid a \in \mathbb{R}\}$, we also obtain that

$$
(1-\varepsilon) L_G^+ \preceq L_{G'}^+ \preceq (1+\varepsilon) L_G^+
$$

using which we further obtain that

$$
\|L_G^+ b - L_{G'}^+ b\|_{L_G}^2 \leq \left( \frac{2}{1-\varepsilon} - 2 \right) b^\top L_{G'}^+ b \leq \frac{2\varepsilon(1+\varepsilon)}{1-\varepsilon} b^\top L_G^+ b \leq 4\varepsilon \|L_G^+ b\|_{L_G}^2 .
$$

Thus, $\|L_G^+ b - L_{G'}^+ b\|_{L_G} \leq 2\sqrt{\varepsilon} \|L_G^+ b\|_{L_G}$. $\qquad\square$

Therefore, if $x$ is a vector such that $\|x - L_{G'} b\|_{L_{G'}} \leq \alpha \|L_{G'}^+ b\|_{L_{G'}}$, obtained using the fast Laplacian solver, then

$$
\begin{aligned}
\|x - L_G^+ b\|_{L_G}^2 &= \|x - L_{G'}^+ b + L_{G'}^+ b - L_G^+ b\|_{L_G}^2 \\
&\leq 2(\|x - L_{G'}^+ b\|_{L_G}^2 + \|L_{G'}^+ b - L_G^+ b\|_{L_G}^2) \\
&\leq \frac{2}{1-\varepsilon} \|x - L_{G'}^+ b\|_{L_G'}^2 + 4\varepsilon \|L_G^+ b\|_{L_G}^2 .
\end{aligned}
$$

Here we used the above theorem and the fact that $L_G \preceq (1/(1-\varepsilon)) L_{G'}$. Now, $\|x - L_{G'}^+ b\|_{L_{G'}}^2 \leq \alpha^2 \|L_{G'}^+ b\|_{L_{G'}}^2$ and $\|L_{G'}^+ b\|_{L_{G'}}^2 = b^\top L_{G'}^+ L_{G'} L_{G'}^+ b = b^\top L_{G'}^+ b \leq (1+\varepsilon) b^\top L_G^+ b \leq (1+\varepsilon) \|L_G^+ b\|_{L_G}^2$, which finally implies that

$$
\|x - L_G^+ b\|_{L_G}^2 \leq \left( \frac{2(1+\varepsilon)^2}{1-\varepsilon} \alpha^2 + 4\varepsilon \right) \|L_G^+ b\|_{L_G}^2 .
$$

Thus, using a $\varepsilon$ spectral sparsifier $G'$ with $m$ edges, we can in time $\widetilde{O}(m \log(1/\varepsilon))$ can obtain a vector $x$ such that $\|x - L_G^+ b\|_{L_G} \leq C\sqrt{\varepsilon} \|L_G^+ b\|_{L_G}$ for a large enough constant $C$.

## D.2 Low-rank Approximation of the Kernel Matrix

We derive algorithms for low-rank approximations of the kernel matrix via KDE queries. We present a algorithm for additive error approximation and compare to prior work for relative error approximation.

We first recall the following two theorems. Let $A_{i,*}$ denote the $i$th row of a matrix $A$.

**Theorem D.8** ((Frieze et al., 2004)). *Let $A \in \mathbb{R}^{n \times m}$ be any matrix. Let $S$ be a sample of $O(k/\varepsilon)$ rows according to a probability distribution $(p_1, \ldots, p_n)$ that satisfies $p_i \geq \Omega(1) \cdot \|A_{i,*}\|_2^2 / \|A\|_F^2$ for every $1 \leq i \leq n$. Then, in time $O(mk/\varepsilon \cdot \mathrm{poly}(k, 1/\varepsilon))$, we can compute from $S$ a matrix $U \in \mathbb{R}^{k \times m}$, that with probability at least $0.99$ satisfies*

$$
\|A - AU^T U\|_F^2 \leq \|A - A_k\|_F^2 + \varepsilon \|A\|_F^2 .
$$

**Theorem D.9** ((Chen & Price, 2017), also see (Indyk et al., 2019)). *There is a randomized algorithm that given matrices $A \in \mathbb{R}^{n \times m}$ and $U \in \mathbb{R}^{k \times m}$, reads only $O(k/\varepsilon)$ columns of $A$, runs in time $O(mk) + \mathrm{poly}(k, 1/\varepsilon)$, and returns $V \in \mathbb{R}^{n \times k}$ that with probability $0.99$ satisfies*

$$
\|A - VU\|_F^2 \leq (1+\varepsilon) \min_{X \in \mathbb{R}^{n \times k}} \|A - X\|_F^2 .
$$

Therefore to compute the low rank approximation, we just need sample from the distribution on rows required by Theorem D.8. We reduce this question to evaluating KDE queries as follows: If $K$ is the kernel matrix, each row of $K$ is the weight of the edges of the corresponding vertex. Therefore, each $p_i$ in the distribution $(p_1, \ldots, p_n)$ is the sum of edge weights *squared* for vertex $x_i$. From vertex queries (Algorithm 4), we know that we can get the degree of each vertex, which is the sum of edge weights. We can extend Algorithm 4 to sample from the sum of *squared* edge weights of each vertex as follows. Consider a kernel $k$ such that there exists an absolute constant $c$ that satisfies $k(x, y)^2 = k(cx, cy)$ for all $x, y$. Such a $c$ exists for the most popular kernels such as the Laplacian, exponential, and Gaussian kernels for which $c = 2, 2$, and $4$ respectively. Thus give our dataset $X$, we simply construct KDE queries for the dataset $X' := cX$. Then by sampling the degrees of the vertices associated with the kernel graph $K'$ of $X'$, we can sample from the distribution required by Theorem D.8 by invoking Algorithm 4 on the dataset $X'$. In particular, using $n$ KDE queries for $X'$, we can get row norm squared values for all rows of our original kernel matrix $K$. We can then sample the rows according to Theorem D.8 and fully construct the rows that are sampled. Altogether, this takes $n$ KDE queries and $O(nk/\varepsilon)$ kernel function evaluations to construct a rank $k$ approximation of $K$; see Algorithm 9.

**Corollary D.10.** *Given a dataset $X$ of size $n$, there exists an algorithm that outputs a rank $k$ matrix $B$ such that*

$$\|K - B\|_F^2 \le \|K - K_k\|_F^2 + \varepsilon \|K\|_F^2$$

*with probability $99/100$, where $K$ is a kernel matrix associated with $X$ based on a Laplacian, exponential, or Gaussian kernel, and $K_k$ is the optimal rank-$k$ approximation of $K$. It uses $n$ KDE queries and $O(nk/\varepsilon \cdot \text{poly}(k, 1/\varepsilon) + nkd/\varepsilon)$ post-processing time.*

We remark that for the application presented in this subsection, we can we can replace 1.1. Indeed, since we only estimate row sums, we only require that the value of a KDE query is at least $\tau$, that is, the average value $\frac{1}{|X|} \sum_{x \in X} k(x, y) \ge \tau$ for a query $y$. Note that via Cauchy Schwartz, this automatically implies a lower bound for the average squared sum:

$$\frac{1}{|X|} \sum_{x \in X} k(x, y)^2 \ge \frac{1}{|X|^2} \left( \sum_{x \in X} k(x, y) \right)^2 \ge \tau^2.$$

---

**Algorithm 9** Additive-error Low-rank Approximation

---

1: Let $c$ be the constant such that $k(x, y)^2 = k(cx, cy)$ for all inputs $x, y$
2: **for** $i = 1$ to $i = n$ **do**
3:     Compute the value $p_i = \sum_{j=1}^n k(cx_i, cx_j)$ using KDE queries for the dataset $cX$
4: **end for**
5: Sample and construct $O(k/\varepsilon)$ rows of $K$ according to probability proportional to $\{p_i\}_{i=1}^n$
6: Compute $U$ from the sample, using Theorem D.8
7: Compute $V$ from the sample, using Theorem D.9
8: **return** Return $V, U$

---

## D.3 Approximating the Spectrum in EMD

In this subsection, we obtain a sublinear time algorithm to approximate the spectrum of the normalized Laplacian associated with the graph whose adjacency matrix is given by the kernel matrix $K$.

The eigenvalues of the Laplacian capture fundamental combinatorial properties of the graph such as community structures at varying scales. See the works (Lee et al., 2012; Louis et al., 2012; Kwok et al., 2013; Czumaj et al., 2015; Gluch et al., 2021), which show that the $j$th eigenvalue of the Laplacian informs us if the graph can be partitioned into $j$ distinct clusters. However, computing a large number of eigenvalues of the Laplacian may not be computationally feasible. Thus, it is desirable to obtain a succinct summary of all eigenvalues, i.e. the spectrum.

Additionally, models of random graphs that aim to describe social or biological networks often times have closed form descriptions of the spectrum for graphs drawn from the model. Borrowing

an example from (Cohen-Steiner et al., 2018), "if the spectrum of random power-law graphs does not closely resemble the spectrum of the Twitter graph, it suggests that a random power-law graph might be a poor model for the Twitter graph." Thus, another application of computing an approximation of the spectrum of eigenvalues is to test the applicability of generative graph models.

Our notion of approximation deals with the Earth mover (EMD) distance.

**Definition D.4** (Earth Mover Distance). *Given two multi-sets of $n$ points in $\mathbb{R}^d$, denoted by $A$ and $B$, the earth-mover distance between $A$ and $B$ is defined as the minimum cost of a perfect matching between the two sets, i.e.*

$$EMD(A, B) = \min_{\pi:A \to B} \sum_{a \in A} \|a - \pi(a)\|_2, \tag{D.1}$$

*where $\pi$ ranges over all one-to-one mappings.*

We can now invoke the algorithm `ApproxSpectralMoment` of (Cohen-Steiner et al., 2018). The algorithm first selects uniformly random vertices of a weighted graph $A$. It then performs a random walk of a specified length $\ell$ starting from the chosen vertex and then counts the number of times the walk returns back to the original vertex. Now Theorem C.7 allows us to perform one step of a random walk using $O(\log n)$ KDE queries. Note that we perform an additional $\tilde{O}(1/\tau)$ of rejection sampling in Algorithm 5 to perfectly sample from the true neighbor distribution. Thus we immediately have the following guarantee:

**Theorem D.11** (Corollary of Theorem 1 in (Cohen-Steiner et al., 2018) and Theorem C.7). *Given a $n \times n$ kernel matrix $K$ and accuracy parameter $\varepsilon \in (0,1)$, let $G$ be the corresponding weighted graph, and let $L_G = I - D^{-1}KD^{-1}$ be the normalized Laplacian, where $D_{i,i} = \sum_j K_{i,j}$. Let $\lambda_1 \geq \lambda_2 \ldots \geq \lambda_n$ be the eigenvalues of $L_G$ and let $\lambda$ be the resulting vector. Then, there exists an algorithm that uses $\tilde{O}\left(\exp\left(1/\varepsilon^2\right)/\tau\right)$ KDE queries and $\exp\left(1/\varepsilon^2\right) \cdot d/\tau$ post-processing time and outputs a vector $\tilde{\lambda}$ such that with probability $99/100$,*

$$EMD\left(\lambda, \tilde{\lambda}\right) \leq \varepsilon.$$

We remark that the bound of $\exp\left(1/\varepsilon^2\right)$ is *independent* of $n$, which is the size of the dataset.

## D.4 First Eigenvalue and Eigenvector Approximation

Our goal is to approximate the top eigenvalue of the kernel matrix and find a vector witnessing this approximation. Our overall algorithm can be split into two steps: first sample a random principal submatrix of the kernel matrix. Under the condition that each row of the $n \times n$ kernel matrix $K$ satisfies that it's sum is at least $n\tau$, we can easily show that it must have a large first eigenvalue and thus prior works on sampling bounds automatically imply the first eigenvalue of the sampled matrix approximates that of $K$. The next step is to use a 'noisy' power method of (Backurs et al., 2021) on the *sampled submatrix*. We note that this step employs a KDE data-structure initialized only on the sampled indices of $K$. The algorithm and details follow.

---

**Algorithm 10** First Eigenvalue and Eigenvector Approximation

---

**Require:** Input dataset $X \subset \mathbb{R}^d$ of size $|X| = n$, precision $\varepsilon > 0$
1: $t \leftarrow O(1/(\varepsilon^2 \tau^2))$
2: $S \leftarrow$ random subset of $[n]$ of size $t$
3: $K_S \leftarrow$ principal submatrix of $K$ on indices in $S$ {Just for notation; we do not initializing $K$ or $K_S$}
4: $\tilde{K} \leftarrow (n/s) \cdot K_S$
5: **Return** the eigenvalue and eigenvector found by Algorithm 1 of (Backurs et al., 2021) (Kernel Noisy Power Method) on $K_S$.

---

We remark that the eigenvector returned by Algorithm 10 will be a *sparse* vector supported only on the coordinates in $S$.

We first state the necessary auxiliary statements needed to prove the guarantees of Algorithm 10.

**Lemma D.12.** *If each row of $K$ satisfies that its sum is at least $n\tau$ for parameter $\tau \in (0,1)$, then the largest eigenvalue of $K$, denoted as $\lambda_1$, satisfies $\lambda_1 \geq n\tau$.*

*Proof.* This follows from looking at the quadratic form $\mathbf{1}^T K \mathbf{1}$ where $\mathbf{1}$ is the vector with all entries equal to 1:

$$\lambda_1 \geq \frac{\mathbf{1}^T K \mathbf{1}}{\mathbf{1}^T \mathbf{1}} \geq \frac{n^2 \tau}{n} = n\tau. \qquad \square$$

We now state the guarantees of Algorithm 1 in (Backurs et al., 2021).

**Theorem D.13** ((Backurs et al., 2021))**.** *Suppose the kernel function for a $m \times m$ kernel matrix $K$ has a KDE data structure with query time $d/(\varepsilon^2 \tau^p)$ (see Table 1). Then Algorithm 1 of (Backurs et al., 2021) returns $\lambda$ such that $\lambda \geq (1-\varepsilon)\lambda_1(K)$ in time $O\left(\frac{dm^{1+p}\log(m/\varepsilon)^{2+p}}{\varepsilon^{7+4p}}\right)$*

Finally, we need the following result on eigenvalues of sampled PSD matrices, proven in (Bhattacharjee et al., 2021).

**Lemma D.14** ((Bhattacharjee et al., 2021))**.** *Let $A \in \mathbb{R}^{n \times n}$ be PSD with $\|A\|_\infty \leq 1$. Let $S \subset [n]$ be a random subset of size $t$ and let $A_{S \times S}$ be the submatrix restricted to columns and rows in $S$ and scaled by $n/s$. Then, for all $i \in [|S|]$, $\lambda_i(A_{S \times S}) = \lambda_i(A) \pm \frac{n}{\sqrt{t}}$.*

We are now ready to prove the guarantees of Algorithm 10.

**Theorem D.15.** *Given a $n \times n$ kernel matrix $K$ admitting a KDE data-structure with query time $d/(\varepsilon^2 \tau^p)$, Algorithm 10 returns $\lambda$ such that $\lambda \geq (1-\varepsilon)\lambda_1(K)$ in total time*

$$\min\left(O\left(\frac{d\log(d/\varepsilon)}{\varepsilon^{4.5}\tau^4}\right), O\left(\frac{d}{\varepsilon^{9+6p}\tau^{2+2p}}\log\left(\frac{1}{\varepsilon\tau}\right)^{2+p}\right)\right).$$

**Remark D.1.** Two remarks are in order. First we recall that the runtime of (Backurs et al., 2021) has a $n^{1+p}$ factor while our bound *has no dependence* on $n$ and is thus a truly sublinear runtime. Second, if we skip the Kernel Noisy Power method step and directly initialize and calculate the top eigenvalue of $K_S$ (using the standard gap independent power method of (Musco & Musco, 2015)), we would get a runtime of $\tilde{O}(d/(\varepsilon^{4.5}\tau^4))$ which has a polynomially better $\varepsilon$ dependence but a worse $\tau$ dependence than the guarantees of Algorithm 10.

*Proof of Theorem D.15.* We first prove the approximation guarantee. By our setting of $t$ and using Lemma D.14, we see that the additive error in approximating the first eigenvalue of $K$ by that of $\tilde{K}$ is at most

$$\frac{n}{\sqrt{t}} \leq \varepsilon\tau n \leq \varepsilon\lambda_1(K),$$

and thus $\lambda_1(\tilde{K}) \geq (1-\varepsilon)\lambda_1(K)$. Then by the guarantees of Theorem D.13, it follows that we find a $1 - \varepsilon$ multiplicative approximation to $\lambda_1(\tilde{K})$ and thus a $1 - O(\varepsilon)$ multiplicative approximation to that of $\lambda_1(K)$.

We now prove the runtime bound. It easily follows from plugging in $m = O(1/(\varepsilon^2\tau^2))$ in Theorem D.13. $\qquad \square$

# E  Graph Applications

In this section, we present our graph applications, including local clustering, spectral clustering, arboricity estimation, and estimating the total weight of triangles.

## E.1 Local Clustering

---

**Algorithm 11** Local $k$-Clustering

---

**Require:** Vertices $u, w$, random walk length $t$
 1: For a given $u$, let $p_u^t$ be the endpoint distribution of a random walk of length $t$ starting at $v$
 2: **if** $\ell_2$ distribution tester outputs $\|p_u^t - p_w^t\|_2 \leq 1/(7n)$ **then**
 3:     **return** $u, w$ are in the same cluster
 4: **end if**
 5: **return** $u, w$ are in different clusters

---

We give a local clustering algorithm on graphs. The advantage of this method is that it is *local* as it allows us to cluster one vertex at a time. This is especially useful in the setting of local clustering where one might not wish to classify all vertices at once or only a small subset of vertices are of interest.

We now present a definition for a clusterable graph that has been an extremely popular model definition in the property testing and sublinear algorithms community (see (Kale & Seshadhri, 2008; Czumaj & Sohler, 2010; Goldreich & Ron, 2011; Czumaj et al., 2015; Chiplunkar et al., 2018; Dey et al., 2019; Gluch et al., 2021) and the references within).

First, we need to define the notion of conductance.

**Definition E.1** (Conductance). *Let $G = (V, E, w)$ be a weighted graph. The conductance of a set $S \subset V$ is defined as*

$$\phi_G(S) = \frac{w(S, S^c)}{\min(w(S), w(S^c))}$$

*where $w(S, S^c)$ denotes the sum of edge weights crossing the cut $(S, S^c)$ and $w(S)$ denotes the sum of (weighted) degrees of vertices in $S$. The conductance of the graph $G$ is then the minimum of $\phi_G(S)$ over all sets $S$:*

$$\phi(G) = \min_S \phi_G(S).$$

**Definition E.2** (Inner/Outer Conductance). *For a subset $U \subseteq V$, we define $\phi(G[U])$ to be the conductance of the induced graph on $U$. $\phi(G[U])$ is also referred to as the inner conductance of $U$. Conversely, $\phi_G(U)$ is refereed to as the outer conductance of $U$.*

**Definition E.3** ($k$-clusterable Graph). *A graph $G$ is $(k, \phi_{in}, \phi_{out})$-clusterable if the following holds: There exists a partition of the vertex set into $h \leq k$ parts $V = \cup_{1 \leq i \leq h} V_i$ such that $\phi(G[V_i]) \geq \phi_{in}$ and $\phi_G(V_i) \leq \phi_{out}$.*

Definition E.3 captures the intuition that one can partition the graph into $h \leq k$ pieces where each piece has a strong cluster structure (captured by $\phi_{in}$) and distinct pieces are separated by sparse cuts (captured by $\phi_{out}$). Note that we are interested in the regime where $\phi_{out}$ is smaller than $\phi_{in}$. We will also assume that each $|V_i| \geq n/\operatorname{poly}(k)$ where we allow for an arbitrary polynomial dependence on $k$. This means that each cluster size is not too small.

Since we are interested in clustering, through this section, we will assume our kernel graph $K$ is $k$-clusterable according to Definition E.3 but we do not know what the partitions are.

The main algorithmic result of this section is that given a $k$-clusterable kernel graph and two vertices $u$ and $w$ that are in parts $V_1$ and $V_2$ respectively of the graph (as defined in Definition E.3), we can efficiently test if $V_1 = V_2$ or $V_1 \neq V_2$. That is, we can efficiently test if $u$ and $w$ belong to the same or distinct clusters. The underlying idea behind the algorithm is that if $u$ and $w$ belong to the same cluster, then random walks starting from these vertices will rapidly mix inside the corresponding cluster. Therefore, random walks in distinct clusters will be substantially different and can be detected using distribution testing. Our algorithm is given in Algorithm 11. The flavor of the algorithm presented is quite standard in property testing literature, see (Czumaj et al., 2015) and (Peng, 2020).

The $\ell_2$ distribution tester we need is a standard result in distribution testing with the following guarantees.

**Theorem E.1** (Theorem 1.2 in (Chan et al., 2014)). *Let $\delta, \xi > 0$ and let $p, q$ be two discrete distributions over a set of size $n$ with $b \geq \max\{\|p\|_2^2, \|q\|_2^2\}$. Let $r \geq c\sqrt{b}\log(1/\delta)/\xi$ for an appropriate constant $c$. There exists $\ell_2$ distribution tester that takes as input $r$ samples from each distribution $p, q$ and accepts the distributions if $\|p - q\|_2^2 \leq \xi$, and rejects the distributions if $\|p - q\|_2^2 \geq 4\xi$ with probability at least $1 - \delta$. The running time of the tester is linear in its sample size.*

We now prove the correctness of Algorithm 11. We note that many arguments from prior works are re-derived in the proof below, rather than stating them in a black box manner, for completeness since our setting is of weighted graphs and the usual setting in literature is unweighted or regular graphs. We first need the following lemmas. Recall that the random walk matrix of an arbitrary weighted graph is given by $M = AD^{-1}$ where $A$ is the adjacency matrix and $D$ is the diagonal degree matrix. The normalized Laplacian matrix $L$ is defined as $L = I - D^{-1/2}AD^{-1/2}$.

Our first result is that vertices in the same well connected cluster of $G$ have a quantitative relationship captured by the eigenvectors of $L$. This is in similar spirit to Lemma 5.3 of (Czumaj et al., 2015) but we must show it holds for weighted graphs arising from kernel matrices whereas (Czumaj et al., 2015) is interested in bounded degree unweighted graphs.

**Lemma E.2.** *Let $\mathbf{v}_i$ be the $i$th eigenvector of the normalized Laplacian of the kernel graph $K$ and let $C$ be any subset such that $\phi(K[C]) \geq \phi_{in}$. Then for any $1 \leq i \leq h$, the following holds:*

$$\sum_{u,v \in C} \left( \frac{\mathbf{v}_i(u)}{\sqrt{w(u)}} - \frac{\mathbf{v}_i(v)}{\sqrt{w(v)}} \right)^2 \lesssim \frac{\phi_{out} n}{\phi_{in}^2 |C| \tau^2}.$$

*Proof.* By Lemma 5.2 in (Czumaj et al., 2015) and Theorem 1.2 in (Lee et al., 2012), we have that $\phi_{in}^2/h^4 \lesssim \lambda_{h+1}$ and $\lambda_i \leq 2\phi_{out}$ for any $1 \leq i \leq h$. Now by the variational principle for eigenvalues (Chung & Graham, 1997), we have

$$\lambda_i = \sum_{(u,v)} \left( \frac{\mathbf{v}_i(u)}{\sqrt{w(u)}} - \frac{\mathbf{v}_i(v)}{\sqrt{w(v)}} \right)^2 w(u, v) \leq 2\phi_{out}.$$

Now let $H = K[C]$. From (Chung & Graham, 1997) and our assumptions on $C$, we have that

$$\text{vol}_H(V_H) \cdot \frac{2 \cdot \sum_{(u,v) \in E_H} \left( \frac{\mathbf{v}_i(u)}{\sqrt{w(u)}} - \frac{\mathbf{v}_i(v)}{\sqrt{w(v)}} \right)^2 w(u,v)}{\sum_{u,v \in V_H} \left( \frac{\mathbf{v}_i(u)}{\sqrt{w(u)}} - \frac{\mathbf{v}_i(v)}{\sqrt{w(v)}} \right)^2 d_H(u)d_H(v)} \geq \lambda_2(H) \geq \frac{\phi_{in}^2}{2},$$

where $\text{vol}_H(V_H)$ denotes the sum of the degrees of vertices in $H$ and $d_H(\cdot)$ denotes the degree in $H$. Note the last step is due to Cheeger's inequality. Combining the preceding result with our earlier derivation, we have

$$\sum_{(u,v) \in E_H} \left( \frac{\mathbf{v}_i(u)}{\sqrt{w(u)}} - \frac{\mathbf{v}_i(v)}{\sqrt{w(v)}} \right)^2 \leq \sum_{(u,v) \in E_K} \left( \frac{\mathbf{v}_i(u)}{\sqrt{w(u)}} - \frac{\mathbf{v}_i(v)}{\sqrt{w(v)}} \right)^2 \leq 2\phi_{out}.$$

This implies that

$$|C|^2 \tau^2 \sum_{(u,v) \in V_H} \left( \frac{\mathbf{v}_i(u)}{\sqrt{w(u)}} - \frac{\mathbf{v}_i(v)}{\sqrt{w(v)}} \right)^2 \leq \sum_{(u,v) \in V_H} \left( \frac{\mathbf{v}_i(u)}{\sqrt{w(u)}} - \frac{\mathbf{v}_i(v)}{\sqrt{w(v)}} \right)^2 d_H(u)d_H(v)$$

$$\lesssim \frac{\phi_{out}\text{vol}_H(V_H)}{\phi_{in}^2}$$

where we have used the fact that all edge weights in $K$ are at least $\tau$. Using the fact that $\text{vol}_H(V_H) \leq |C|n$, it follows that

$$\sum_{(u,v) \in V_H} \left( \frac{\mathbf{v}_i(u)}{\sqrt{w(u)}} - \frac{\mathbf{v}_i(v)}{\sqrt{w(v)}} \right)^2 \lesssim \frac{\phi_{out} n}{\phi_{in}^2 |C| \tau^2},$$

as desired. □

The second result states that vertices in the same well-connected cluster have similar random walk distributions. This is again the analogue of Lemma 4.2 in (Czumaj et al., 2015) but we must show it holds for weighted graphs.

**Lemma E.3.** *Let $0 < \beta < 1/2$. If graph $K$ is $(k, \phi_{in}, \phi_{out})$-clusterable, and $C \subseteq V$ is any subset such that $|C| \geq n/\operatorname{poly}(k)$ and $\phi(K[C]) \geq \phi_{in}$. There exists a constant $c = c(\beta) > 0$ and $c' = c'(\beta, k)$ such that for any $t \geq c \log n/\phi_{in}^2$, $\phi_{out} \leq c'\phi_{in}^2$, there exists a subset $\widetilde{C} \subseteq C$ satisfying $vol(\widetilde{C}) \geq (1-\beta)vol(C)$ such that for any $u, v \in \widetilde{C}$, the following holds:*

$$\|p_u^t - p_v^t\|_2^2 \leq \frac{1}{8n}.$$

*Proof.* Let $\mathbf{v}_1, \cdots, \mathbf{v}_n$ denote the eigenvectors of $L$ with eigenvalues $\lambda_1, \cdots, \lambda_n$ in non-decreasing order. We know that the eigenvalues of $M$ are given by $1 - \lambda_i$ with corresponding eigenvalues $\mathbf{y}_i = D^{1/2}\mathbf{v}_i$. The vector $p_u^t$ is the vector $1_u$ with a one value in the $u$th coordinate applied to $M^t$. Write

$$1_u = \sum_i \alpha_i y_i = \sum_i \alpha_i D^{-1/2}\mathbf{v}_i.$$

Taking the innerproduct of $1_u$ with $D^{-1/2}\mathbf{v}_i$ tells us that $\alpha_i = \mathbf{v}_i(u)/\sqrt{w(u)}$. Thus,

$$p_u^t - p_v^t = \sum_{i=1}^n \frac{\mathbf{v}_i(u)}{\sqrt{w(u)}}(1 - \lambda_i)^t \mathbf{y}_i - \sum_{i=1}^n \frac{\mathbf{v}_i(v)}{\sqrt{w(v)}}(1 - \lambda_i)^t \mathbf{y}_i$$

$$= D^{1/2}\sum_{i=1}^n \mathbf{v}_i \left( \frac{\mathbf{v}_i(u)}{\sqrt{w(u)}} - \frac{\mathbf{v}_i(v)}{\sqrt{w(v)}} \right)(1 - \lambda_i)^t.$$

This means that

$$\|p_u^t - p_v^t\|_2 \leq \|D^{1/2}\| \left\| \sum_{i=1}^n \mathbf{v}_i \left( \frac{\mathbf{v}_i(u)}{\sqrt{w(u)}} - \frac{\mathbf{v}_i(v)}{\sqrt{w(v)}} \right)(1 - \lambda_i)^t \right\|_2.$$

Since the $\mathbf{v}_i$'s are orthogonal, we know that

$$\left\| \sum_{i=1}^n \mathbf{v}_i \left( \frac{\mathbf{v}_i(u)}{\sqrt{w(u)}} - \frac{\mathbf{v}_i(v)}{\sqrt{w(v)}} \right)(1 - \lambda_i)^t \right\|_2^2 \leq \sum_{i=1}^n \left( \frac{\mathbf{v}_i(u)}{\sqrt{w(u)}} - \frac{\mathbf{v}_i(v)}{\sqrt{w(v)}} \right)^2 (1 - \lambda_i)^{2t}.$$

Now the rest of the proof follows from Lemma 4.2 in (Czumaj et al., 2015). In particular, it tells us that in the above summation, each of the terms for $1 \leq i \leq h$ can be bounded by $\lesssim \frac{\phi_{out} n}{\beta |C|^3 \phi_{in}^2}$ whereas the rest of the sum can be bounded by $1/\operatorname{poly}(n)$ for sufficiently large $\operatorname{poly}(n)$ by adjusting the constant in front of $t$. Our choice for $|C| \geq n/\operatorname{poly}(k)$ imply that the overall sum is bounded by $\lesssim \frac{\phi_{out} \operatorname{poly}(k)}{\beta n^2 \phi_{in}^2}$. Since $\|D^{1/2}\|^2 \leq n$, and $\phi_{out} \leq c'\phi_{in}^2$, we have that $\|p_u^t - p_v^t\|_2^2 \leq 1/(8n)$, as desired. $\qquad\square$

Our next goal is to show that vertices from different well-connected partitions have very different random walk endpoint distributions. The argument we borrow is from (Czumaj & Sohler, 2010).

**Lemma E.4.** *Let $G$ be a $(k, \phi_{in}, \phi_{out})$-clusterable graph with parts $V = \cup_{1 \leq i \leq h} V_i$. There exists a constant $c > 0$ such that if $t \cdot \phi_{out} \leq c\varepsilon$, then there exists a subset $V_1' \subseteq V_1$ satisfying $vol(V_1') \geq (1-\varepsilon)vol(V_1)$ such that a $t$-step random walk from any vertex in $V_1'$ does not leave $V_1$ with probability $1 - \varepsilon$.*

*Proof.* We first bound the probability that the random walks always stay in their respective clusters. Consider a fixed partition $V_1$; the same arguments apply for any partition. Let $G'$ be the graph with the same vertex set as $K$ but with only the following edges: edges among vertices in $V_1$ and edges from vertices in $V_1$ to $V \setminus V_1$. Consider a random walk on $G'$ of length $t$ with the initial vertex $u'$ chosen from the stationary distribution of $G'$, i.e., the distribution that chooses each vertex in $G'$ with probability proportional to its weight. Let $Y_i$ denote the indicator random variable for the event that

the $i$th vertex of the random walk is in $V \setminus V_1$. Since we are simulating the stationary distribution, we have that

$$\Pr[Y_i = 1] = \frac{w(V_1, V \setminus V_1)}{w(G')}$$

where $w$ is the weight of edges in the original graph $K$. By linearity of expectations, the number of vertices that land in $V \setminus V_1$ is

$$\mathbb{E}\left[\sum_{i=1}^{t} Y_i\right] = (t+1)\frac{w(V_1, V \setminus V_1)}{w(G')} \lesssim t\phi_{out}$$

due to our requirement of $\phi_{out}$. Therefore by Markov's inequality, the probability that any vertex in $V \setminus V_1$ is ever visited is $\lesssim t\phi_{out}$.

We now move our random walk back to the original graph $K$. The preceding calculation implies that the probability that an $t$ step random walk in $K$ starting at a vertex chosen at random from $V_1$ according to the stationary distribution will remain in $V_1$ with probability at least $1 - \phi_{out}t$. If $\phi_{out} \lesssim \varepsilon/t$, then we know that the random walk stays in $V_1$ with probability at least $1 - O(\varepsilon)$ so there must be a set of vertices $V' \subseteq V_1$ of at least $1 - O(\varepsilon)$ fraction of the total volume of $V_1$ such that a random walk starting from a vertex in $V'$ remains in $V_1$ with probability at least $1 - O(\varepsilon)$. $\square$

We can now prove the correctness of Algorithm 11.

**Theorem E.5.** *Let $K$ be a $(k, \phi_{in}, \phi_{out})$-clusterable kernel graph with parts $V = \cup_{1 \le i \le h} V_i$. Let $U, W$ be one of (not necessarily distinct) partitions $V_i$. Let $u, w$ be randomly chosen vertices in partitions $U$ and $W$ with probability proportional to their degrees. There exists $c = c(\varepsilon, k)$ such that if $\phi_{out} \le c\phi_{in}^2/\log n$, then with probability at least $1 - \varepsilon$, if $U = W$ then Algorithm 11 returns that $u$ and $w$ are in the same cluster and if $U \neq W$, Algorithm 11 returns that $u$ and $w$ are in different clusters. The algorithm requires $O(\sqrt{nk/(\varepsilon\tau)}\log(1/\varepsilon))$ random walks of length $t \ge c\log n/\phi_{in}^2$.*

*Proof.* We first consider the case that $U \neq W$. From Lemma E.4, we know that there are 'non-escaping' subsets $U'$ and $W'$ of $U$ and $W$ respectively such that vertices $u, w$ from $U'$ and $W'$ respectively don't leave $U$ and $W$ with probability $1 - \varepsilon$. Conditioning on $u$ and $w$ being in those subsets, we have that with probability $1 - O(\varepsilon)$, $p_u^t$ and $p_v^t$ will be disjointly supported and thus, $\|p_u^t - p_v^t\|_2^2 = \|p_u^t\|_2^2 + \|p_v^t\|_2^2 \ge 2/n$.

Now if $U = W$, we know from Lemma E.3 that $\|p_u^t - p_v^t\|_2^2 \le 1/(8n)$ if we condition on $u$ and $v$ coming from the large volume subset of $U$.

Finally, we need one last ingredient. Lemma 4.3 in (Czumaj et al., 2015) readily implies that there exists a $V' \subseteq V$ satisfying $\text{vol}(V') \ge (1-\varepsilon)\text{vol}(V)$ such that $\|p_u^t\|_2^2 \le 2k/(\varepsilon\tau^2 n)$. Now we can set $\xi = 1/(7n)$ and $b = 2k/(\varepsilon\tau^2 n)$ in Theorem E.1, which tells us that $r = O(\sqrt{nk/(\varepsilon\tau)}\log(1/\varepsilon))$ samples of the distributions $p_u^t$ and $p_w^t$ suffice to distinguish the cases $\|p_u^t - p_v^t\|_2^2 \ge 2/n$ or $\|p_u^t - p_v^t\|_2^2 \le 1/(8n)$, i.e., $r$ samples allow us to determine if $U = W$ or $U \neq W$, conditioned on a $1 - O(\varepsilon)$ probability event. $\square$

It is straightforward to translate the requirements of Theorem E.5 in terms of the number of KDE queries required. Note that since we only take random walks of length $O(\log n/\phi_{in}^2)$, we can just reduce the total variation distance from the distribution we sample our walks from and the true random walk distribution appropriately in Theorem C.7. Alternatively, we can perform rejection sampling as stated in the proof of Theorem C.5.

**Corollary E.6.** *Algorithm 11 and Theorem E.5 require $\widetilde{O}(c(k, \varepsilon)\sqrt{nk/\varepsilon} \cdot 1/(\tau^{1.5}\phi_{in}^2))$ KDE queries (via calls to Algorithm 7, which performs random walks) as well as the same bound for post-processing time.*

## E.2 Spectral Clustering

We present applications to spectral clustering. In data science, spectral clustering is often the following clustering procedure: (a) compute $k$ eigenvalues of the Laplacian matrix in order, (b) perform $k$-means clustering on the Laplacian eigenvector embeddings of the vertices.

The theory behind spectral clustering relies on the fact that the Lapalacian eigenvectors are effective in representing the cluster structure of the underlying graph. We refer the reader to (Von Luxburg, 2007) and references within for more information. For our application to spectral clustering, we show that a spectral sparsifier, for example one computed from the prior sections, also preserves the cluster structure of the graph.

Next we define a model of a "weakly clusterable" graph. Intuitively our model says that a graph is $k$-weakly clusterable if its vertex set can be partitioned into $k$ 'well-connected' pieces separated by sparse cuts in between. Furthermore, this definition captures the notion of a well-defined cluster structure without which performing spectral clustering is meaningless. Note that this notion is less stringent that the definitions of clusterable graphs commonly used in the property testing literature which additionally require each piece to be well-connected internally, see Definition E.3.

**Definition E.4** (Weakly clusterable Graph). *A graph is $(k, \phi_{out})$-clusterable if the following holds: There exists a partition of the vertex set into $h \leq k$ parts $V = \cup_{1 \leq i \leq h} V_i$ such that $\phi_G(V_i) \leq \phi_{out}$.*

We now prove the following result that says spectral sparsification preserves cluster structure according to Definition E.4. We first remark that the spectral sparsifier obtained in the previous section is a *cut sparsifier* as well. Recall that a cut sparsifier is a subgraph that preserves the values across all cuts up to relative error $1 \pm \varepsilon$. The implication follows immediately by noting that cuts are induced by quadratic forms on the Laplacian matrix using $\{-1, 1\}^n$ vectors.

**Theorem E.7.** *Let $G$ be $(k, \phi_{out})$-clusterable and let $G'$ be a cut sparsifier for $G$. Then $G'$ is $(k, (1 \pm \varepsilon)\phi_{out})$-clusterable.*

*Proof.* Let $V_i$ be one of the $h \leq k$ vertex partitions of $G$. Consider the conductance of $V_i$ defined in Definition E.1. The numerator represents the value of a cut separating $V_i$ and each term in the denominator is the sum of the degrees of single vertices. Both values are appropriate cuts in the graph. Since $G'$ is a cut sparsifier, this implies that both the numerator and denominator are preserved up to a $1 \pm \varepsilon$ factor and thus, the entire ratio is also preserved up to a $1 \pm O(\varepsilon)$ factor. □

Theorem E.7 implies that the cluster structure of the sparsified graph $G'$ is approximately identical to that of $G$. Thus, we can be confident that the spectral clustering procedure described at the beginning of the section would perform equally as well on $G'$ as it would have on $G$. Indeed, we verify this empirically in Section 2.

Furthermore, spectral clustering requires us to compute the first $k$ eigenvectors of the Laplacian matrix. Since our sparsifier has few edges, we can use Theorem 1 of (Musco & Musco, 2015), which says (a variant of) the power method can quickly find good approximations Laplacian eigenvectors if the matrix is sparse.

**Theorem E.8** (Corollary of Theorem 1 in (Musco & Musco, 2015) and Theorem D.1). *Let $L$ be the Laplacian matrix of the sparsifier computed in Theorem D.1. Let $u_1, \cdots, u_k$ be the first $k$ eigenvectors of $L$. Using Theorem 1 of (Musco & Musco, 2015), we can find $k$ vectors $v_1, \cdots, v_k$ in time $\widetilde{O}\left(\frac{kn \log n}{\tau^2 \varepsilon^{2.5}}\right)$ such that with probability $99/100$,*

$$|u_i^T L u_i - v_i^T L v_i| \leq \varepsilon \lambda_{k+1}^2$$

*for all $i \in [k]$.*

### E.3 Arboricity Estimation

---

**Algorithm 12** Arboricity Estimation

---

1: $\Delta = \max_{e,e' \in E} \frac{w(e)}{w(e')}$
2: $m \leftarrow O\left(\frac{n\Delta \log n}{\varepsilon^2}\right), G' \leftarrow \emptyset$
3: **for** $i = 1$ to $i = m$ **do**
4:     Sample an edge $e$ with probability $p_e = \frac{\widehat{w_e}}{\sum \widehat{w_e}}$, where $\widehat{w_e} \in [w_e, 2w_e]$
5:     Add $e$ to $G'$ with weight $\frac{1}{mp_e}$
6: **end for**
7: **return** $\max_{U \subseteq V} d(G'_U)$

---

We now apply our algorithmic building blocks to the task of arboricity estimation. Consider a weighted graph $G = (V, E, w)$. Let $G_U$ be an induced subgraph of $G$ on the subset of nodes $U$. The density of $G_U$ is defined as

$$d(G_U) := \frac{w(E(G_U))}{|U|}$$

where $w(E(G_U))$ is the sum of the edge weights of $G_U$. The arboricity of $G$ is defined as

$$\alpha := \max_{U \subseteq V} d(G_U).$$

The arboricity measures the density of the densest subgraph in a graph. Intuitively, it informs if there is a strong cluster structure among some subset of the vertices of $G$. Therefore, it is an important primitive in the analysis of massive graphs with applications ranging from community detection in social networks, spam link identification, and many more; see (Lee et al., 2010) for a survey of applications and algorithmic results.

Although polynomial time algorithms exist, we are interested in efficiently approximating the value of $\alpha$ using the building blocks of Section C. Inspired by the unweighted version of the arboricity estimation algorithm from (McGregor et al., 2015), we first prove the following result.

**Theorem E.9.** *Let $U' = \arg\max_U d(G'_U)$ and let $G'$ be the output of Algorithm 12. Then with probability at least $1 - 1/\operatorname{poly}(n)$,*

$$(1 - \varepsilon)\alpha \leq d(G'_{U'}) \leq (1 + \varepsilon)\alpha.$$

*Algorithm 12 uses $m = \widetilde{O}(n \log n/(\varepsilon^2 \tau))$ KDE queries and $O(mn)$ post-processing time.*

*Proof.* Let $U$ be an arbitrary set of $k$ nodes, let $W = \sum_{e \in E} w(e)$, and let $W_U = W \cdot d(G_U)$ Since $G$ has weight $W$, then the arboricity $\alpha$ satisfies $\alpha \geq \frac{W}{n}$, so that

$$m \geq \frac{\log n}{\alpha \Delta \varepsilon^2}.$$

Let $X_i$ be the random variable denoting the contribution of the $i$-th sample to weight of the edges in $G'_U$ and observe that $\mathbb{E}[X_i] = \frac{W}{m} \cdot d(G_U)$ so that $\mathbb{E}[X] = W \cdot d(G_U)$, for $X = \sum_{i=1}^m$. Similarly, we have

$$\mathbb{E}[X_i^2] = \sum_{e=(u,v),u,v,\in U} p_e w_e^2 \cdot \frac{1}{p_e^2 m^2} = \sum_{e=(u,v),u,v,\in U} \frac{W w_e}{m^2} = \frac{W \cdot W_U}{m^2}.$$

Since $m = \frac{Cn\Delta \log n}{\varepsilon^2}$ for an absolute constant $C > 0$, then

$$\mathbb{E}[X_i^2] \leq \frac{k\varepsilon^2 \alpha^2}{Cm \log^2 n}$$

and

$$\sum_{i=1}^m \mathbb{E}[X_i^2] \leq \frac{k\varepsilon^2 \alpha^2}{Cm \log^2 n}.$$

We also have $X_i - \mathbb{E}[X_i] \leq \frac{\alpha\varepsilon^2}{Cm\log^2 n}$. Thus by Bernstein's inequality for sufficiently large $C$,

$$\Pr\left[d(G'_U) \geq \frac{\alpha}{10}\right] \leq n^{-10k},$$

for $d(G_U) \leq \frac{\alpha}{60}$ and

$$\Pr\left[|d(G'_U) - d(G_U)| \geq \frac{\varepsilon\alpha}{10}\right] \leq 2n^{-10k},$$

for $d(G_U) > \frac{\alpha}{60}$.

Since there are $\binom{n}{k} \leq n^k$ subsets of $V$ with size $k$, then by a union bound, we have that with probability at least $1 - 3n^{-9k}$, both

$$d(G'_U) \leq \frac{\alpha}{10}$$

for all subsets $U$ with $d(G_U) \leq \frac{\alpha}{60}$ and

$$(1-\varepsilon)d(G_U) \leq d(G'_U) \leq (1+\varepsilon)d(G_U)$$

for all subsets $U$ with $d(G_U) > \frac{\alpha}{60}$.

Hence for a set $U^*$ such that $d(U^*) = \alpha$, we have $d(G_{U^*}) \geq (1-\varepsilon)\alpha$ so that $d(G_{U'}) \geq d(G_{U^*}) \geq (1-\varepsilon)\alpha$, where $U' = \arg\max_{U \subseteq V} d(G'_U)$. Thus with high probability, we have that

$$(1-\varepsilon)\alpha \leq d(G_{U'}) \leq (1+\varepsilon)\alpha,$$

as desired. $\qquad\square$

To estimate the arboricity of the input graph $G$, it then suffices Theorem E.9 to compute the arboricity of the subsampled graph $G'$ output by Algorithm 12. This can be efficiently achieved by running an offline algorithm such as (Charikar, 2000), which requires solving a linear program on $m$ variables, where $m$ is the number of edges of the input graph. Thus our subsampling procedure serves as a preprocessing step that ultimately significantly improves the overall runtime.

## E.4 Computing the Total Weight of Triangles

---
**Algorithm 13** Weighted Triangle Counting

---
1: Let $R \subseteq E$ be a random set of $O\left(\frac{m\sqrt{w_G}w_{\max}^{3/2}}{\varepsilon^2 w_T}\right)$ edges
2: $s \leftarrow O\left(\frac{w_G^2 w_{\max}^3 m^2}{w_{\min}}\right)$
3: For $v \in V$, $g(v) := \sum_{(u,v)\in E} w(u,v)$
4: For $(u,v) \in E$, $g(u,v) := \min(w(u), w(v))$
5: **for** $i = 1$ to $i = s$ **do**
6:     Sample $e \in R$ with probability $\frac{g(e)}{\sum_{e\in E} g(e)}$
7:     Let $e = (x, y)$ with $x \prec y$
8:     Sample neighbor $z$ of $x$ with probability $\frac{w(x)}{g(e)}$
9:     **if** $(x, y, z)$ is a triangle assigned to $u$ **then**
10:       $\chi_i \leftarrow 1$
11:     **else**
12:       $\chi_i \leftarrow 0$
13:     **end if**
14: **end for**
15: **return** $\frac{m}{|R|s} \sum_{i=1}^{s} \chi_i$

---

We apply the tools developed in prior section to counting the number of weighted triangles of a kernel graph. Counting triangles is a fundamental graph algorithm task that has been explored in numerous models and settings, including streaming algorithms, fine-grained complexity, distributed

shared-memory and MapReduce to name a few (Seshadri et al., 2013; Atserias et al., 2008; Bera & Chakrabarti, 2017; Kolountzakis et al., 2010; Chen et al., 2022). Applications include discovering motifs in protein interaction networks (Milo et al., 2002), understanding social networks (Foucault Welles et al., 2010), and evaluating large graph models (Leskovec et al., 2008); see the survey (Al Hasan & Dave, 2018) for further information.

We define the weight of a triangle as the product of its edges. This definition is natural since it generalizes the case were the edges have integer lengths. In this case, an edge can be thought of as multiple parallel edges. The number of triangles on any set of three vertices must account for all the parallel edge combinations. The product definition just extends this to the case of arbitrary real non-negative weights. This definition has also been used in definitions of clustering-coefficient for weighted graphs (Kalna & Higham, 2006; Li et al., 2007; Antoniou & Tsompa, 2008).

Note that there is an alternate definition for the weight of a triangle in weighted graphs, which is just the sum of edge weights. In the case of kernel graphs, this is not an interesting definition since we can approximately compute the sum of all degrees using $n$ KDE queries and divide by 3 to get an accurate approximation.

**Definition E.5.** *Let $G = (V, E, w)$ with $w : E \to \mathbb{R}^{\geq 0}$ be a weighted graph. Given a triangle $(x, y, z) \subset E$, we define its weight as*

$$w_{(x,y,z)} = w(x, y) \cdot w(y, z) \cdot w(x, z),$$

*where we abuse notation by defining $w(x, y) := w((x, y))$.*

For this definition, we present the following modified algorithm from (Eden et al., 2017), which considers the problem in unweighted graphs in a different model of computing. See Remark B.5 for comparison.

**Theorem E.10.** *There exists an algorithm that makes $\widetilde{O}\left(\frac{m\sqrt{w_G}w_{\max}^{3/2}}{w_T \cdot \varepsilon^2}\right)$ KDE queries and the same bound for post-processing time and with probability at least $\frac{2}{3}$, outputs a $(1 \pm \varepsilon)$-approximation to the total weight $w_T$ of the triangles in the kernel graph.*

*Proof.* Given a graph $G = (V, E)$, let $|V| = n$, $|E| = m$, $\sum_{e \in E} w(e) = w_G$, $T$ be the number of triangle in $G$, and $w_T$ be the sum of the weighted triangles in $G$, where the weight of a triangle $(x, y, z) \subset E$, is the product of the weights of its edges

$$w_{(x,y,z)} = w(x, y) \cdot w(y, z) \cdot w(x, z).$$

For a vertex $v \in V$, let $w(v) = \sum_{e \in E, e=(u,v), u \in V} w(e)$, so that we have an ordering on the vertex set $V$ by $u \prec v$ if and only if either $w(u) \leq w(v)$ or $w(u) = w(v)$ and $u$ appears $v$ in the dictionary ordering of the vertices. For each edge $e = (u, v)$, we assign to $e$ all triangle $(u, v, w)$ such that $u \prec v \prec w$. Let $W_e$ denote the weight of the triangles assigned to $e$.

Suppose, by way of contradiction, there exists $e \in E$ with $W_e > \sqrt{w_G}w_{\max}^{3/2}$. Since each triangle can contribute at most $w_{\max}^3$ weight to $W_e$, then more than $\sqrt{w_G}w^{-3/2}$ triangles must be assigned to $e$. Then there must be more than $\sqrt{w_G}w^{-3/2}$ vertices with weight at least $\sqrt{w_G}w_{\max}^{3/2}$, which contradicts the fact that the graph $G$ has weight $w_G$. Thus, we have that $W_e \leq \sqrt{w_G}w_{\max}^{3/2}$ for all $e \in E$. Moreover, we have that $\sum_{e \in E} W_e = w_T$ so that

$$\mathbb{E}_{e \in E}[W_e] = \frac{w_T}{m}$$

and

$$\mathbb{E}_{e \in E}[W_e^2] \leq \frac{w_T \sqrt{w_G}w_{\max}^{3/2}}{m}.$$

By Chebyshev's inequality, it suffices to sample a set $R$ with

$$|R| = O\left(\frac{m\sqrt{w_G}w_{\max}^{3/2}}{\varepsilon^2 w_T}\right)$$

edges uniformly at random, so that

$$\Pr\left[\sum_{e\in R} W_e \in (1\pm\varepsilon)|R|\cdot\frac{w_T}{m}\right] \ge 0.99.$$

For each vertex $v \in V$, let $g(v) = \sum_{(u,v)\in E} w(u,v)$ and for each edge $e = (u,v)$, let $g(e) = \min(w(u), w(v))$. We write $g(R) = \sum_{e\in R} g(e)$. Now for each $i \in [|R|]$, we have

$$\mathbb{E}[\chi_i] = \sum_{e\in R} \frac{W_e}{g(R)} = \frac{W_R}{g(R)}, \qquad \mathbb{E}[\chi_i^2] \le 1.$$

Hence by Bernstein bounds, there exists a constant $C > 0$ such that it suffices to repeat the procedure $\frac{C}{\varepsilon^2}\cdot\frac{\cdot g(R)}{W_R}$ times to get a $(1\pm\varepsilon)$-approximation of $\frac{W_R}{g(R)}$ with probability at least $2/3$. We have that $W_R \ge \frac{w_T}{2m}\cdot|R|$ and $\mathbb{E}[g(R)] \le \sqrt{w_G}w_{\max}^{3/2}\cdot|R|$ so that

$$\frac{g(R)}{W_R} \le \frac{2m\sqrt{w_G}w_{\max}^{3/2}}{w_T}. \qquad \square$$

## F    Omitted Experimental Results

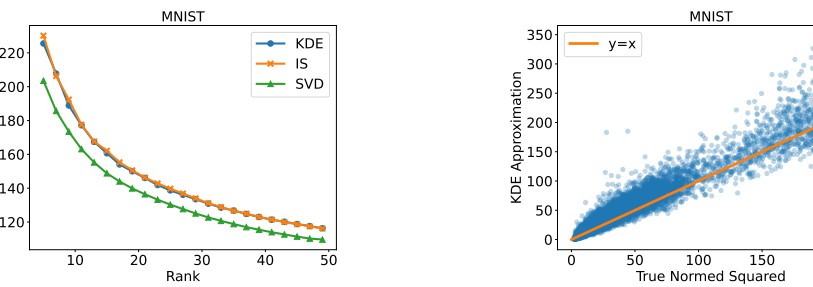

(a) Rank versus Error for Low-rank Approximation    (b) Real vs Approximate Row Norm Squared Values

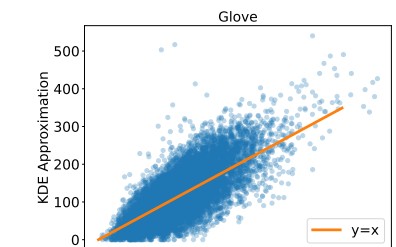

(c) Rank versus Error for Low-rank Approximation    (d) Real vs Approximate Row Norm Sqaured Values

Figure 2: Figures for low rank approximation experiments.

**Parameter settings.**    For low-rank approximation datasets, we choose the bandwidth value $\sigma$ according to the choice made in prior experiments, in particular the values given in (Backurs et al., 2019). There, $\sigma$ is chosen according to the popular median distance rule; see their experimental section for further information. For our clustering experiments, we pick the value of $\sigma$, which results in spectral clustering (running on the full kernel matrix) successfully clustering the input.

**Datasets for spectral sparsification and clustering.** For spectral sparsification and clustering, we use construct two synthetic datasets, which are challenging for other clustering method such as $k$-means clustering[2]. The first dataset denoted as 'Nested' consists of $5,000$ points, equally split among the origin and a circle of radius $1$. The two natural clusters are the points at the origin and the points on the circle. Since one cluster is contained in the convex hull of the other, a method like $k$-means clustering will not be able to separate the two clusters, but it is known that spectral clustering can. Our second dataset, labeled 'Rings', is an even more challenging clustering dataset. We consider two tori in three dimensions that pass through the interior hole of each other, i.e., they interlock. The 'small' radius of each tori is $5$ while the 'large' radius is $100$. Our dataset consists of $2500$ points uniformly distributed on the two tori; see Figure 3b. Note that our focus is not to compare the efficacy of various clustering methods, which is done in other prior works (e.g., see footnote 2). Rather, we show that spectral clustering itself can be optimized in terms of runtime, space usage, and the number of kernel evaluations performed via our algorithms.

**Evaluation metrics for spectral clustering and sparsification.** For spectral sparsification and clustering, we compare the accuracy of our method to the clustering solution when run on the full initialized kernel matrix.

Note that prior works such as (Backurs et al., 2019; 2021) have used use the number of kernel evaluations performed (i.e., how many entries of $K$ are computed) as a measure of computational cost. While this is a software and architecture independent basis of comparison, which is unaffected by access to specialized libraries or hardware (e.g., SIMD, GPU), it is of interest to go beyond this measure. Indeed, we use this measure as well as other important metrics as space usage and runtime as points of comparison.

**Spectral sparsification and clustering results.** Our algorithm consists of running the spectral sparsification algorithm of Theorem D.1 (Algorithm 8) and computing the first two eigenvectors of the normalized Laplacian of the resulting sparse graph. We then run $k$-means clustering on the computed Laplacian embedding for $k = 2$. As noted above, we use two datasets that pose challenges for traditional clustering methods such as $k$-means clustering. The Nested dataset is shown in Figure 3a. We sampled $3 \cdot 10^5$ many edges, which is $2.5\%$ of total edges. Figure 4a shows the Laplacian embedding of the sampled graph based on the first two eigenvectors. The colors of the red and blue points correspond to their cluster in Figure 3a as identified by running $k$-means clustering on the Laplacian embedding. The orange crosses are the points that the spectral clustering method failed to correctly classify. These are only $23$ points, which represent a $0.5\%$ of total points. Furthermore, Figure 4a shows that the Laplacian embedding of the sampled graph is able to embed the two clusters into distinct and disjoint regions. Note that the total space savings of the sampled graph over storing the entire graph is **41x**. In terms of the time taken, the iterative SVD method used to calculate the Laplacian eigenvectors took $0.18$ seconds on the sparse graph whereas the same method took $0.81$ seconds on the entire graph. This is a **4.5x** factor reduction.

We recorded qualitatively similar results for the rings dataset. Figure 3b shows a plot of the dataset. We sampled $10^5$ many edges for the approximation, which represents a $3.3\%$ of total edges for form the sparse graph. The Laplacian embedding of the sparse graph is shown in Figure 4b. In this case, the embedding constructed from the sparse graph was able to separate the two rings into disjoint regions perfectly. The time taken for computing the Laplacian eigenvectors for the sparse graph was $0.08$ seconds whereas it took $0.27$ seconds for the full dense matrix.

# G   Auxiliary Inequalities

**Theorem G.1** (Bernstein's inequality). *Let $X_1, \ldots, X_n$ be independent random variables such that $\mathbb{E}[X_i^2] < \infty$ and $X_i \geq 0$ for all $i \in [n]$. Let $X = \sum_i X_i$ and $\gamma > 0$. Then*

$$\Pr\left[X \leq \mathbb{E}[X] - \gamma\right] \leq \exp\left(\frac{-\gamma^2}{2\sum_i \mathbb{E}[X_i^2]}\right).$$

---

[2]For example, see `https://scikit-learn.org/stable/auto_examples/cluster/plot_cluster_comparison.html`.

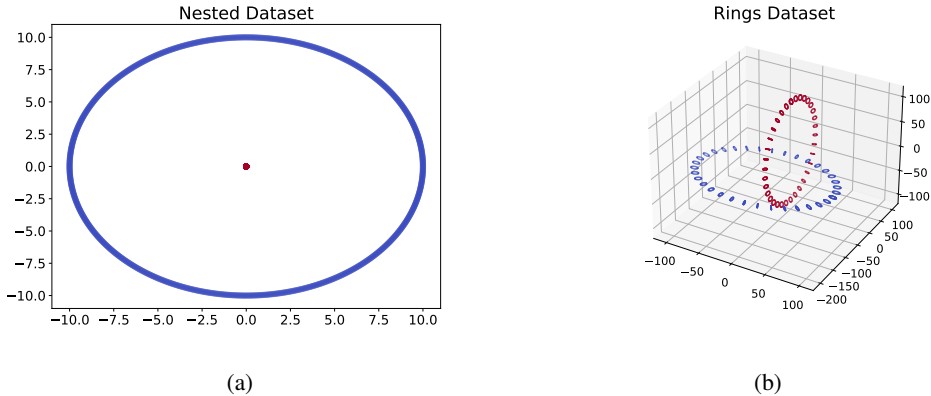

Figure 3: (a) Nested Dataset, (b) Rings Dataset

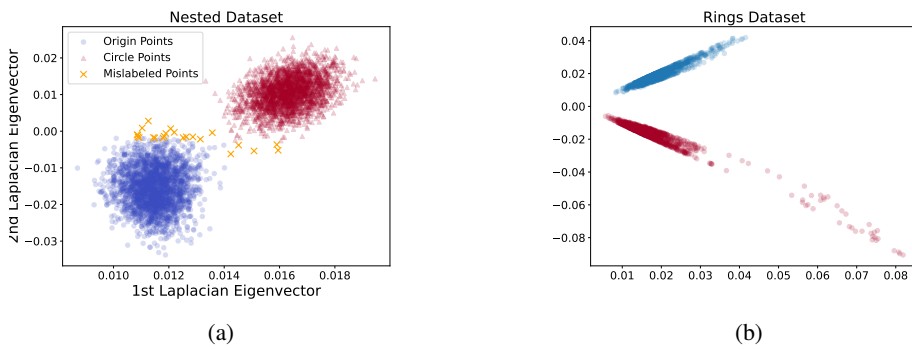

Figure 4: Spectral embedding of sparsified graph for (a) Nested dataset and (b) Rings dataset, respectively.

If $X_i - \mathbb{E}[X_i] \leq \Delta$ for all $i$, then for $\sigma_i^2 = \mathbb{E}[X_i^2] - \mathbb{E}[X_i]^2$,

$$\Pr\left[X \geq \mathbb{E}[X] + \gamma\right] \leq \exp\left(\frac{-\gamma^2}{2\sum_i \sigma_i^2 + 2\gamma\Delta/3}\right).$$

