# OpenReview forum: "Subquadratic Algorithms for Kernel Matrices via Kernel Density Estimation"
_ICLR.cc/2023/Conference — ICLR 2023 notable top 25%_

### Official Review · Reviewer_dkUW · 2022-10-23

**Confidence:** 3
**Correctness:** 4
**Technical Novelty And Significance:** 3
**Empirical Novelty And Significance:** 2
**Recommendation:** 8

**Clarity, Quality, Novelty And Reproducibility:**

Clarity:
I feel the paper is written exceptionally well. Tries to summarize theoretical findings in an intuitive way.

Quality:
The work begins with a core idea of using KDE queries and explores how a hist of different problems can be solved. It also seems to comprehensively theoretically contrast with existing results. I feel the technical contributions are strong.



**Strength And Weaknesses:**

Strengths:
1. The theoretical results in the theorems as well as the remarks are interesting. They also seem to compare well wrt. existing works.
2. The writeup is very well organized by summarizing the theoretically results and proofs in an intuitive way. I appreciate the discussion in section 1.2

Weakness:
1. Though I understand the contribution is theoretical, the simulations seem  to be too restricitive. Only low rank based approximations are compared. Also, here, it would have been nice to see a comparison between time vs approximation, without which it is hard to evaluate the improvement.

**Summary Of The Paper:**

 Using KDE queries (as black box), the paper proposes to effciently solve various gram matrix related problems such as spectral approximation/sparsification, simulating random walks, weighted sampling, etc. In each case, a theoretical bound on the approximation/quality along with no. KDE queries is presented. Simulation results on low rank approximation are provided.

**Summary Of The Review:**

In view of the strong theory and nice presentation I tend to recommend an acceptance though I feel simulations section needs a heavy upgrading.

--
After seeing the rebuttal and other reviews, I increase my score.

---

> ### Author Response · Authors · 2022-11-11
> **Thank you to reviewer dkUW.**
>
> > Though I understand the contribution is theoretical, the simulations seem to be too restricitive.
>
> Our empirical evaluation serves as a proof of concept that our queries, which we constructed, are efficient and easy to implement in practice. The queries form the basis of all our algorithmic results and our low rank approximation results show they can indeed be implemented efficiently.
>
> > Only low rank based approximations are compared.
>
> We believe the reviewer may have missed additional experiments given in the paper. As stated in the empirical evaluation section in the main body, we also present experimental evaluations for Spectral sparsification and clustering in Appendix F.  Evaluating all of the numerous algorithms which can be used in black box manner in conjunction with our queries would drastically increase the length of the paper, so we leave it for future work. Any practical implementation of such algorithms can be easily combined with our queries to obtain an algorithm for kernel matrices
>
> > Also, here, it would have been nice to see a comparison between time vs approximation, without which it is hard to evaluate the improvement.
>
> We note that the empirical section of the paper already discusses run time versus approximation. To summarize, we see that for other low rank approximation methods, such SVD and IS described in the paper,  the time to initialize the KDE matrix *dominates* the runtime. The sampling method presented in the paper does not need to initialize the matrix which makes the method scalable compared to the benchmarks in terms of runtime, space, and also the number of kernel evaluations. Thus as stated in the main body, we see at least a 2.9x factor decrease in the runtime.
>
> The full time values for the MNIST dataset are also given in https://postimg.cc/kBKJLfvj (anonymous image repository) and the plot qualitatively mirrors the times stated in the main body which are stated for the rank=50 case. We also highlight that in Figure 1(a) and 1(c), the green curve represents the error of SVD, which obtains the optimal error given a fixed rank. We see that our algorithm only incurs a small factor increase in error compared to the optimal error at the benefit of much faster runtime. For a fixed rank, it is possible to tune our low-rank approximation algorithm to obtain a smaller error (up to the error obtained by SVD). Theorem 1.5 dictates that we can set a smaller additive error by decreasing the error parameter epsilon at the cost of increased runtime. The main parameter we can tune is the number of rows sampled in Theorem D.8, which is used to prove Theorem 1.5.  In the figure in https://postimg.cc/NKwmDPY6, we fixed a rank parameter (rank = 20) and iterated over the parameters of the algorithm to obtain a runtime vs approximation plot. The plot exactly depicts the guarantees of Theorem 1.5 as the approximation error is smaller for larger runtimes.
>
> We emphasize that other metrics such as the number of kernel valuations are also important as it represents a software and architecture free measure “unaffected by access to specialized libraries (e.g., BLAS, MATLAB) or hardware (e.g., SIMD, GPU)” [1]. Further as stated in [1], “this is important with linear algebraic operations, which behave very differently in different environments, resulting in artifacts when measuring runtimes.” For this important measure, our method used close to an order of magnitude less number of evaluations compared to the baselines which require the full kernel matrix initialization.
>
> [1]: Faster Kernel Matrix Algebra via Density Estimation. Arturs Backurs, Piotr Indyk, Cameron Musco, Tal Wagner (ICML ‘21).
>
> We hope we have addressed your concerns and we would be happy to engage in further discussions.

---

> > ### Author Response · Authors · 2022-11-16
> > **Follow up to reveiwer dkUW.**
> >
> > Dear reveiwer dkUW,
> >
> > Did we address all your concerns satisfactorily? If your concerns have not been resolved, could you please let us know which concerns were not sufficiently addressed so that we have a chance to respond before the November 18 deadline?
> >
> > Many thanks, The authors

---

> > > ### Comment · Reviewer_dkUW · 2022-11-18
> > > **Increased score**
> > >
> > > Thanks for the clarifications. I increased my score accordingly.

---

> > > > ### Author Response · Authors · 2022-11-18
> > > > **Thank you.**
> > > >
> > > > Thank you to reviewer dkUW for the valuable discussion!

---

### Official Review · Reviewer_vw4c · 2022-10-24

**Confidence:** 3
**Correctness:** 3
**Technical Novelty And Significance:** 4
**Empirical Novelty And Significance:** 2
**Recommendation:** 8

**Clarity, Quality, Novelty And Reproducibility:**

For the most part the content is clear, and I will provide instances where this is not the case (which I found) in the following section. Overall the paper appears to include a substantial amount of novel and quality work. A conscientious reader should be able to reproduce and even extend the methods and experiments given in the paper, with sufficient effort.

**Strength And Weaknesses:**

Strengths:
- The paper includes a fairly extensive coverage of the important operations required in popular machine learning applications.
- The content is comprehensive enough that a reader can conceivably extend the algorithms given to any (most?) operations not covered explicitly in the paper.
- The presentation is reasonably good overall, but I will provide some comments on this going forward.
- The approach taken in the paper is a natural application of the KDE queries to the problems covered, which makes much of the content intuitive and easier than it would otherwise be to follow.

Weaknesses:
- There are some issues with clarity (I will provide some examples in the following)
- I have some, what I believe to be important qualms with the way in which the content is presented. In particular, although there is no apparent incorrectness in the paper, the statements are arguably overblown in respect of their practical relevance. For example, as my closest connections to the content in the paper are from the points of view of spectral clustering and non-parametric smoothing using kernels, it is, in my experience, extremely rare that the scaling parameter (\sigma in the paper) used in determining the kernel weights/similarities does not depend at least implicitly on the data set size (n). For example:
-- For most non-parametric smoothing problems one has \sigma ~ scale(X) n^(-1/(4+d)), where scale(X) is a measure of the scale of the data set. Using this, and for example the Gaussian kernel, one has \tau ~ exp(-c*n^(2/(4+d))) for c = diam(X)^2/2scale(X)^2, where diam(X) is the diameter of the data set.
 NB: I see that the authors mention that some literature states that often the smallest element in the Gram matrix is a fixed constant, however that does not seem to be consistent with the kernels suggested in the paper unless the data distribution is compact and the scaling parameter does not tend to zero. In addition, I imagine that in general for good performance the bound on tau needs to be very small, and then the question is "For what sort of data set size do we actually start to realise a substantial improvement in run-time while still maintaining accuracy?"


Let me clarify that I in no way think this invalidates the proposed approach, but I think it is a very important point which is worth discussing and, at the very least noting in the paper.

**Summary Of The Paper:**

The primary focus of the paper is in the use of KDE queries to speed up many of the primitive operations applied to kernel matrices (Gram matrices) for the purpose of speeding up approximate executions of important linear-algebraic and graph theoretic operations/computations. The fundamental idea is that KDE queries allow for the approximation of the sums of kernel evaluations between an entire data set and a given query point in time independent of data set size and pre-processing which is linear in the data set size*.
 Algorithms are given for how these primitive operations are subsequently applied to important applications in linear algebra and computational graph theory.
 The paper concludes with an experiment illustrating the potential of the proposed method for the ubiquitous Singular Value Decomposition operation.

**Summary Of The Review:**

Overall the paper is reasonably well written and includes potentially influential and important methodology in very relevant areas in machine learning. The theory is comprehensive enough to likely even allow readers to fairly easily extend the general method to operations/problems not explicitly in the scope of the paper.

My main concern relates to the fact that in many practical situations, the important "parameter" tau depends on sample size or likely is very small in practice, but this point seems to be almost glossed over in the paper (as well as the related literature). I think the contribution is significant enough that such a suppression (I do not intend to imply that it is deliberate) is unnecessary, and it is important to clarify/discuss where this is and is not an important factor. We see in the experiment that the performance is good FOR THE GIVEN GRAM MATRIX, however it may be that the scaling used may have oversmoothed the structure in the data to the extent that much of the local information is lost. Again, I do not mean to imply this is the case, only that as a reader one does not know.

Below I will conclude with some smaller questions/clarifications/suggestions/issues:
- Using the term sub-linear suggests to a reader, in my opinion, sublinear in the data set size and not sublinear in the size of the Gram matrix. This is unnecessary.
- Definition 1.1 does not seem to be correctly stated. Surely it should read that the query point y satisfies k(x, y) \geq \tau for all x \in X
- Theorem 1.3 requires setting/defining \epsilon
- Theorem 1.4 needs either to state that v is unit norm, or the LHS in the inequality should be divided by ||v||^2.
- Sometimes one sees "... with probability a/b..." and other times "... with probability d%...". Why the inconsistency?
- Below Theorem 1.5: \omega has not been defined
- The statement of Theorem 1.6 is rather clumsy, and could be made clearer. Also, \epsilon features in the number of queries without being defined and also not present in any of the other terms. I presume it is somehow hidden in a probability somewhere, but the statement is "with certainty" and no probability is mentioned.
- In the paragraph below Theorem 1.6, I presume the authors mean they compute the top few eigenVECTORS of the Laplacian
- The notation of the edge-vertex incident matrix is quite confusing. Should the subscript x_i not be i? Also, although not ambiguous, subscripting with the edge is not mathematically precise. Is there a way to maybe define H as a map rather than a matrix?
 Further the expression  pi >= (1-\epsilon)||r_i||^2 = 2(1-\epsilon)k(x_i, x_j) is confusing since index j has not been defined. I presume first you set j and then can sample the rows with such probability?
- Am I totally out of date by thinking the Nystrom approximation is a relevant alternative? How does the empirical performance on the experiment in Section 2 compare when using this ubiquitous approximation method?
- The authors say that in the figure in the experiment, all the points in the true and approximate squared row norms lie close to the y=x line. This is very questionable, especially in the Glove data. Is it not important that lots of the approximate row norms are essentially zero when their actual values are far from that?

---

> ### Author Response · Authors · 2022-11-11
> **Thank you to reviewer vw4c. Response # 1**
>
> > Tau does not depend at least implicitly on the data set size. In addition, I imagine that in general for good performance the bound on tau needs to be very small, and then the question is "For what sort of data set size do we actually start to realise a substantial improvement in run-time while still maintaining accuracy?"
>
> We do not assume that tau, the smallest entry in the kernel matrix, is constant. Rather, we are parameterizing all of our algorithms in terms of tau. This is a natural parameterization for the following reason. If tau is very small, then the design of non-trivial algorithms becomes provably impossible for many natural problems. For example, Theorem D.4 proved in the appendix implies that if tau is polynomially small, then we require super linear time to compute the spectral sparsifier for the kernel graph. The paper [1] presents other lower bounds for problems on kernel matrices which also hold when the entries are sufficiently (polynomially) small. Therefore, parametrizing the runtime in terms of tau allows us to design non trivial algos which can be broadly applied in practice, even if tau is small in practice. Indeed, this is the viewpoint taken by recent works, such as those cited in the related works, which also parameterize algorithms in terms of the entries of the kernel matrix for the sake of algorithm design. Furthermore in practice, parameters are set so that *most* entries of the kernel matrix are not too small, [2, 4 and references within] which suggests that our algorithms can be broadly applied.
>
> Lastly we note that two of our algorithm applications, additive error low rank approximation and approximating the top eigenvector of the kernel matrix, which are two of our most fundamental applications, only rely on the much milder assumptions that the *average* entry in every row is at least tau. Indeed for the top eigenvector result, given access to a KDE data structure, we give an algorithm with runtime *independent* of the size of the dataset. The best prior result is the main result of the paper [3] which has a super linear in n runtime.
>
> [1] ​​On the Fine-Grained Complexity of Empirical Risk Minimization: Kernel Methods and Neural Networks. Arturs Backurs, Piotr Indyk, Ludwig Schmidt. (NeurIPS ‘17).
>
> [2] DEANN: Speeding up Kernel-Density Estimation using Approximate Nearest Neighbor Search. Matti Karppa, Martin Aumuller, Rasmus Pagh. (AISTATS ‘22).
>
> [3] Faster Kernel Matrix Algebra via Density Estimation. Arturs Backurs, Piotr Indyk, Cameron Musco. (ICML ‘21).
>
> [4] Rehashing Kernel Evaluation in High Dimensions. Paris Siminelakis, Kexin Rong, Peter Bailis, Moses Charikar, Philip Levis. (ICML '19).
>
> > Using the term sub-linear suggests to a reader, in my opinion, sublinear in the data set size and not sublinear in the size of the Gram matrix. This is unnecessary.
>
> The term “sublinear” has been used in past work in the context of matrix algorithms whose runtime is sublinear in the number of matrix entries (see [1- 5] and references within). However, for the sake of clarity, we replaced “sublinear” with “subquadratic” in the paper, whenever appropriate.
>
> [1] Sublinear time algorithms for metrics space problems. Piotr Indyk. STOC ‘99
>
> [2] Sublinear time approximation of the cost of a metric k-nearest neighbor graph. Artur Czumaj and Christian Sohler. SODA ‘20
>
> [3] Faster Kernel Matrix Algebra via Density Estimation. Arturs Backurs, Piotr Indyk, Cameron Musco. (ICML ‘21).
>
> [4]  Testing of clustering. Noga Alon, Seannie Dar, Michal Parnas, Dana Ron. SIAM Journal on Discrete Mathematics.
>
> [5] Czumaj, Artur, and Christian Sohler. "Sublinear-time algorithms." Property testing. Springer, Berlin, Heidelberg, 2010. 41-64. (Survey article with additional references)
>
> > Definition 1.1 does not seem to be correctly stated. Surely it should read that the query point y satisfies k(x, y) \geq \tau for all x \in X
>
> We believe the definition as currently written (and in the original submission) is correct as stated. Please let us know if there is any misunderstanding!
>
> > Theorem 1.3 requires setting/defining \epsilon
>
> We apologize and have included further details in the updated version. $\epsilon$ refers to the accuracy parameter. See Theorem D.11 for the formal version of the theorem.
>
> >Theorem 1.4 needs either to state that v is unit norm, or the LHS in the inequality should be divided by ||v||^2.
>
> Yes v is a unit vector. We have updated the text.
>
> > Below Theorem 1.5: \omega has not been defined
>
> $\omega$ is the matrix multiplication constant. We will update for clarity.

---

> > ### Author Response · Authors · 2022-11-11
> > **Response # 2**
> >
> > > The statement of Theorem 1.6 is rather clumsy
> >
> > We apologize for the confusion. The statement has been updated and we refer to Theorem E.5 for the formal version of the theorem for full details.
> >
> > > In the paragraph below Theorem 1.6, I presume the authors mean they compute the top few eigenVECTORS of the Laplacian
> >
> > We can "compute the top few eigenvalues of the Laplacian matrix" and "...the corresponding approximate eigenvectors are used in spectral clustering". That is, our algorithm can compute both the top few eigenvalues and corresponding approximate eigenvectors.
> >
> > > The notation of the edge-vertex incident matrix is quite confusing. Should the subscript x_i not be i? Also, although not ambiguous, subscripting with the edge is not mathematically precise. Is there a way to maybe define H as a map rather than a matrix? Further the expression pi >= (1-\epsilon)||r_i||^2 = 2(1-\epsilon)k(x_i, x_j) is confusing since index j has not been defined. I presume first you set j and then can sample the rows with such probability?
> >
> > We apologize for the notation that may be confusing. H is a (n choose 2) by n matrix where we have a row for every possible edge $e = (x_i, x_j).$ For the row corresponding to edge e, there is an entry in the x_i column with value $\sqrt{k(x_i, x_j)}$ and an entry in the $x_j$ column with value $- \sqrt{k(x_i, x_j)}$. Thus the norm squared of this row is $2k(x_i, x_j)$. This is expanded upon in paragraph after equation (1.1).  We also think the reviewer for pointing out a typo in the definition of the row norms of H in Section 1.2 which has been updated with changes highlighted in blue.
> >
> > > Am I totally out of date by thinking the Nystrom approximation is a relevant alternative? How does the empirical performance on the experiment in Section 2 compare when using this ubiquitous approximation method?
> >
> > We agree that Nystrom approximation is a relevant and practical low rank approximation algorithm. However for the empirical section, we are focused on methods with provable guarantees. Nystrom methods are based on sampling the columns or rows of the kernel matrix and the classical Nystrom method uses uniform sampling. However, it is known that uniform sampling does not give good theoretical guarantees. For example, as stated in [1], consider a dataset where points fall into several clusters with one cluster much larger than the rest. Uniform sampling will tend to oversample “landmarks” used in the Nystrom method overwhelmingly from the large cluster while undersampling or possibly missing smaller but still important clusters. Thus as argued in [1] “Approximation of the kernel matrix and learning performance (e.g. classification accuracy) will decline as a result.” On the other hand, sampling with row norm squared has provable guarantees [2].
> >
> > Note that sampling by the so-called “leverage scores” in Nystrom method does have provable guarantees. However, approximating these sampling probabilities without initializing the entire matrix is quite complicated and could lead to a large number of kernel evaluations. Thus, sampling according to row norm squared presents the best of both worlds: it’s easy to implement while having provable guarantees. We leave it as an interesting future work to compare the benefits of both provable methods and popular heuristics.
> >
> > [1]: Recursive Sampling for the Nystrom Method. Cameron Musco, Christopher Musco. (NeurIPS ‘17).
> >
> > [2]: Fast Monte Carlo Algorithms For Matrices Iii: Computing A Compressed Approximate Matrix Decomposition. Petros Drineas , Ravi Kannan , Michael W. Mahoney. (Siam Journal Of Computing).
> >
> >
> > > The authors say that in the figure in the experiment, all the points in the true and approximate squared row norms lie close to the y=x line. This is very questionable, especially in the Glove data. Is it not important that lots of the approximate row norms are essentially zero when their actual values are far from that?
> >
> > Figures 1(b) and 1(d) show a strong correlation between the true approximate row norms and the approximated value. It is especially important to sample the rows with higher norms with larger probabilities as omitting them leads to a larger variance. We see that our approximations are accurate for the rows which have the largest norms.
> >
> > We hope we have addressed your concerns and we would be happy to engage in further discussions.

---

> > > ### Comment · Reviewer_vw4c · 2022-11-11
> > > **Replies**
> > >
> > > Regarding eigenvalues/eigenvectors:
> > > - The paper currently reads "We obtain an algorithm for approximately
> > > computing the top few eigenvalues of the Laplacian matrix, which is one of the main bottlenecks
> > > in spectral clustering in practice, with subquadratic runtime." As far as I am aware one does not need the eigenvalues themselves for spectral clustering and only the eigenvectors. I am aware that from a practical perspective if getting the eigenvalues leads to the eigenvectors then it doesn't matter, however the bottleneck in spectral clustering is the recovery of the vectors and not values. That is all my comment was trying to say.
> > >
> > > Regarding Nystrom:
> > > - I appreciate your argument. However, as you mention in response to one of my other comments, some recent work suggests that one only requires "most" entries in the Gram matrix to be at least \tau, and so your work may therefore be broadly applicable. However, your theoretical results do not hold in those instances, even if in practice the performance will be good. Comparing with something like the Nystrom is therefore surely reasonable since it is tried and tested and even without theoretical guarantees still provides a practical comparison to your proposal. And let me just stop quickly to say that I suspect your algrorithms likely WILL be broadly accurate often even when the theory does not hold, and that I do like the work. If your method gets as good performance as Nystrom (or better) AND has theoretical guarantees, then it is defensibly better overall
> > > - Just out of interest, you mention leverage scores in Nystrom variants with provable guarantees. Are these leverage scores which can be approximated by your method(s)? If so, does that mean applying your algorithms to non-uniform sampling for Nystrom for overall approximation guarantees is a direction?
> > >
> > > Regarding row norm approximations:
> > > - Correlation doesn't guarantee that all points are accurate. Also, it seems intuitively that it is the smaller row-sums which are often the more important ones on which to achieve accuracy. When looking at probability sampling we surely want the RELATIVE error to be small, since otherwise we could be sampling rows with very small row sums many orders of magnitude too frequently?

---

> > > > ### Author Response · Authors · 2022-11-16
> > > > **Response to reviewer.**
> > > >
> > > > > Regarding eigenvalues/eigenvectors
> > > >
> > > > Thank you for the comment. We agree that our current wording may unintentionally sound like we are claiming that the eigenvalue approximation is the bottleneck for spectral clustering. We have changed the wording to reduce ambiguity.
> > > >
> > > > > Regarding Nystrom
> > > >
> > > > To the best of our knowledge, we are not aware of directly approximating the leverage scores of a kernel matrix using KDE queries, but it is an interesting future direction, as is comparing the benefits of Nystrom versus other approaches with theoretical guarantees. We remark that in some sense, we are simulating leverage score sampling due to oversampling by a factor of $\tau$ with respect to squared row norm sampling in the spectral sparsification section (see the paragraph “Importance Sampling for the edge-vertex incidence matrix and the kernel matrix” and Section D.1). However this introduces a factor of $1/\tau$ in the number of samples used.
> > > >
> > > > > Regarding row norm approximations
> > > >
> > > > In the theoretical guarantees, oversampling a row never hurts in the sense that as long as the probability $p_i$ for sampling the $i$th row $r_i$ of the matrix $A$ satisfies $p_i \ge c \|r_i\|_2^2/\|A\|_F^2$ for a constant c, the guarantees of the theorem holds. Furthermore, we note that the large row norms contribute the most variance. It turns out that the error of algorithms which sample rows $r_i$ with probability $p_i$ is proportional to $\sum_i \frac{\|r_i\|_2^4}{p_i}$ which leads to the choice of $p_i$ being proportional to $\|r_i\|_2^2$. This derivation shows that rows with larger norms have a correspondingly larger contribution to the error (for ex. see slide 9 in [1]. The calculation is also implicit in the proof of Lemma 2 in [2] which is the citation for Theorem D.8).
> > > > However we do acknowledge the reviewer's points that we are sampling from in our experiment are not exactly equal to the true norms, but are highly correlated enough to work well in practice. This naturally leads to the question of exploring other algorithms which do not necessarily have strict theoretical guarantees, but work well in practice. We believe this to be an important question which is out of scope for the experiments of the paper. Our goal is to give a proof of concept that our queries, which we constructed, are efficient and easy to implement in practice.
> > > >
> > > > [1] https://www.epfl.ch/labs/anchp/wp-content/uploads/2018/10/lecture5-slides.pdf
> > > > [2] https://www.math.cmu.edu/~af1p/Texfiles/SVD.pdf

---

> > > > > ### Comment · Reviewer_vw4c · 2022-11-17
> > > > > **Thanks for responses**
> > > > >
> > > > > Thanks a lot for your responses.
> > > > >
> > > > > Just to reiterate, I am not questioning the theory, only that the assertion that the row norms are approximated will is a questionable statement. However, as you say, the subsequent application to the low-rank approximation yields an accurate result. My concern about whether or not the settings in the kernel make the Gram matrix one which actually yields useful analysis, e.g., spectral clustering solutions, remains. Not to suggest this has been done intentionally, but \sigma can always be set very large so that there is little variation in the values of the Gram matrix and \tau is very large, meaning your accuracy will be extremely high, but the kernel has smoothed over any interesting structure in the data.

---

> ### Comment · Reviewer_vw4c · 2022-11-17
> **Thanks and final**
>
> Thanks to the authors for their comments and engagement. I am overall satisfied by the comments and adjustments and think the paper is a good contribution.

---

> > ### Author Response · Authors · 2022-11-17
> > **Response to reviewer.**
> >
> > Thank you as well for the thorough engagement which has made the paper better!

---

### Official Review · Reviewer_Eddv · 2022-10-27

**Confidence:** 4
**Correctness:** 4
**Technical Novelty And Significance:** 3
**Empirical Novelty And Significance:** Not applicable
**Recommendation:** 8

**Clarity, Quality, Novelty And Reproducibility:**

Overall, the paper is fairly well-written and the results are very interesting. I think this would shed the light on how the KDE problem can be embedded into various applications while preserving the sublinear time efficiency.


Small typos:

- In Theorem 1.8, “$\alpha$” should be “$\alpha_G$”
- In Theorem 1.8, does “$\tilde{O}$” mean hiding logarithmic dependency? I think it does not need to use “$\tilde{O}$” or remove $\log n$.
- In page 20, “line 8 or line 11” should be “line 7 or line 10”
- In line 12 of Algorithm 8, “graph G” should be “graph G’”
- In page 24, the $\kappa$ should be defined before it is mentioned


**Strength And Weaknesses:**

Strength:

- The KDE is a fundamental problem in a wide range of areas and this work enhances the impact of the KDE by exploring various practical problems. Moreover, all problems preserve sublinear runtimes that are from the efficient KDE.
- For efficient usage of the KDE, these works propose a multi-level KDE, which combines a tree structure. This incurs a logarithmic overhead hence sublinear-time advantage is preserved.
- All the proposed methods use the KDE as a black-box approach, so further improvements on the KDE will automatically improve that of these tasks.
- All applications have rigorous guarantees, which make the usage of the KDE reliable and much stronger.

Weakness:

- Due to the page limit, the main paper contains the overview only and all key contents are in the appendix.
- Some of the applications are somewhat straightforward and trivial. So, I think it might be better if the authors reduce all applications to some specific ones, and make the main contributions thin that are not straightforward.


**Summary Of The Paper:**

This work studies broad linear algebraic and graph processing applications using the kernel density estimation (KDE) framework which provides sublinear-time algorithms in the number of vertices. A core approach is to sample vertices and edges with respect to the (kernel-based) weights using the KDE in sublinear time and construct a sparsified Laplacian, which guarantees a spectral approximation. Such a (spectrally) well-approximated Laplacian provides rigorous guarantees of downstream problems, such as low-rank approximation, graph clustering, and so on. Experimental results on low-rank approximation and graph sparsification are provided.

**Summary Of The Review:**

The results of this paper are strong and impactful. The paper is well-written and very clear.

---

> ### Author Response · Authors · 2022-11-11
> **Thank you to reveiwer Eddv.**
>
> > So, I think it might be better if the authors reduce all applications to some specific ones, and make the main contributions things that are not straightforward.
>
> We thank the reviewer for the suggestion. We believe that the breadth of applications of our method is worth highlighting, even if some applications are quite simple. However, we will rethink how to present the results so that simple applications do not obscure the more complex ones.
>
>
> > Small typos
>
> The typos have been fixed in the updated version.

---

### Author Response · Authors · 2022-11-11
**Thank you to all reviewers.**

We thank the reviewers for their valuable feedback. Answers are given in a response to each review. We have uploaded a revised version of the paper, with newly added segments marked in blue for the convenience of the reviewers (we will remove the coloring in the final version).

---

### Decision · Program_Chairs · 2023-01-20

**Decision:**

Accept: notable-top-25%

**Justification For Why Not Higher Score:**



**Justification For Why Not Lower Score:**



**Metareview: Summary, Strengths And Weaknesses:**

The focus of the submission is scaling up the approximate computation of various tasks which rely on Gram matrices associated to reproducing kernels, with particular interest in linear algebra and graph processing (such as low rank approximation, computation of dominant eigenvectors / eigenvalues, spectral sparsification, triangle or arboricity estimation; see Table 2). The authors present a novel set of algorithms which have sublinear runtime w.r.t. the number of elements in the Gram matrix (Theorem 1.2-1.9), relying on recent advances in KDE (kernel density estimation). The theoretical insights are accompanied with numerical demonstrations on both synthetic and real-world benchmarks.

Kernel methods are among the most flexible techniques with numerous fundamental applications; providing accelerated schemes in this area is of primary importance to the ICLR and data science community. As it was unequivocally expressed by the reviewers, the paper is clearly written, with substantial amount of novel high-quality contributions.

**Note From Pc:**

if the above contains the word "oral" or "spotlight" please see: "oral" presentation means -> notable-top-5% and "spotlight" means -> notable-top-25%. As stated in our emails, we are disassociating presentation type from AC recommendations